



Investigation on the abnormal quasi-two day wave activities during
sudden stratospheric warming period of January 2006
Sheng-Yang Gu[1,2]*, Xiankang Dou[1,2,3], Dora Pancheva[4]
[1]CAS Key Laboratory of Geospace Environment, Department of Geophysics and
Planetary Science, University of Science and Technology of China, Hefei, Anhui,
China
[2]Mengcheng National Geophysical Observatory, School of Earth and Space Sciences,
University of Science and Technology of China, Hefei, Anhui, China
[3]Wuhan University, Wuhan, China
[4]National Institute of Geophysics, Geodesy and Geography, BAS, Sofia, Bulgaria
*Corresponding Author: Sheng-Yang Gu (gsy@ustc.edu.cn)



**Abstract**

The quasi-two day wave (QTDW) during austral summer period usually
coincides with sudden stratospheric warming (SSW) event in the winter hemisphere,
while the influences of SSW on QTDW are not totally understood. In this work, the
anomalous QTDW activities during the major SSW period of January 2006 are further
investigated on the basis of hourly Navy Operational Global Atmospheric Prediction
System-Advanced Level Physics High Altitude (NOGAPS-ALPHA) reanalysis
dataset. Strong westward QTDW with zonal wave number 2 (W2) is identified
besides the conventionally dominant mode of zonal wave number 3 (W3). Meanwhile,
the W3 peaks with an extremely short period of ~42 hours. Compared with January
2005 with no evident SSW, we found that the zonal mean zonal wind in the summer
mesosphere is enhanced during 2006. The enhanced summer easterly sustains critical
layers for W2 and short-period W3 QTDWs with larger phase speed, which facilitate
their amplification through wave-mean flow interaction. The stronger summer
easterly also provides stronger barotropic/baroclinic instabilities and thus larger
forcing for the amplification of QTDW. The inter-hemispheric coupling induced by
strong winter stratospheric planetary wave activities during SSW period is most likely
responsible for the enhancement of summer easterly. Besides, we found that the
nonlinear interaction between W3 QTDW and the wave number 1 stationary planetary
wave (SPW1) may also contribute to the source of W2 at middle and low latitudes in
the mesosphere.





## 1. Introduction

The temperature and wind fields in the mesopause region exhibit strong oscillation with a period of several days, of which the Quasi-Two-Day wave (QTDW) is the most frequently reported planetary wave (*Palo et al.*, 2007; *Limpasuvan and Wu*, 2009; *McCormack et al.*, 2009; *Pedatella and Forbes*, 2012; *Yue et al.*, 2012; *Chang et al.*, 2014; *Siskind and McCormack*, 2014; *Guharay et al.*, 2015; *Lilienthal and Jacobi*, 2015; *Madhavi et al.*, 2015; *Gu et al.*, 2016a; *Pancheva et al.*, 2016; *Wang et al.*, 2017). There are both eastward and westward QTDWs with different zonal wave numbers (*McCormack et al.*, 2014; *Gu et al.*, 2016b; *Pancheva et al.*, 2016). The eastward QTDWs are usually found to exist in the winter hemisphere (*Sandford et al.*, 2008; *Gu et al.*, 2017), while the westward modes tend to be summer phenomena that peak shortly after the solstice (*Pancheva et al.*, 2004; *Tunbridge et al.*, 2011). In the southern hemisphere, the westward QTDWs show maximum amplitude during January/February at middle and low latitudes (*Limpasuvan and Wu*, 2003; *Palo et al.*, 2007; *Gu et al.*, 2013a). In the northern hemisphere, the QTDW peaks intermittently from June to August at middle latitudes (*McCormack et al.*, 2014; *Gu et al.*, 2016b; *Pancheva et al.*, 2016). Generally, the QTDW activities during the austral summer period are much stronger than those during boreal summer period and thus have received more attention.

The propagation and amplification of planetary waves are intimately related to the background zonal wind. As for the QTDW, it has been shown that the baroclinic/barotropic instability of the summer easterly jet is an important source for its amplification (*Chang et al.*, 2011; *Yue et al.*, 2012). The Eliassen-Palm (EP) flux





associated with QTDW grows dramatically near its critical layer (where the
background wind equals its phase speed), which indicates the energy transportation
from mean flow. The Advanced Level Physics High Altitude version of the Navy
Operational Global Atmospheric Prediction System (NOGAPS-ALPHA) reanalysis
dataset shows that the inter-annual variations of the QTDW during boreal summer
period are dependent on the strength of the summer easterly. A stronger summer
easterly provides larger forcing for its amplification (*McCormack et al.*, 2014).
Recently, *Gu et al.* (2016a) found that the strength of the summer easterly is also
responsible for the selective amplification of QTDWs with different zonal wave
numbers. The westward zonal wave number 2 (W2) QTDW peaks with a stronger
summer easterly than the westward zonal wave number 3 (W3) mode. This is because
a stronger summer easterly can sustain a critical layer for QTDW with larger phase
speed (e.g., W2), and the amplification of QTDW occurs more easily at the unstable
region with a critical layer (*Liu et al.*, 2004; *McCormack et al.*, 2014).
Sudden Stratospheric Warmings (SSWs) occur in the winter stratosphere, and are
most frequently observed during boreal winter period (December-February). The
zonal mean temperature at 10 hPa and 60ºN can increase by tens of Kelvin in one or
two weeks during a SSW event. It is called a major SSW if the westerly wind at 10
hPa and 60ºN reverses, while the winter westerly is slowed down but does not become
easterly during a minor SSW. It is generally accepted that the westward forcing from
the rapid amplification of planetary waves is responsible for the wind deceleration or
reversal in the winter stratosphere (*Matsuno*, 1971; *Liu and Roble*, 2002).



Interestingly, the occurrence of SSW in the northern hemisphere winter stratosphere
usually coincides with the temporal variation of the QTDW in the summer
mesosphere. Nevertheless, their influence on each other has not been totally
understood yet.

Evidence has been found for inter-hemispheric coupling during a SSW event,

which may have significant modulation on summer easterly jet and thus the
amplification of planetary waves. *Karlsson et al.* (2007) showed that the noctilucent
cloud in the summer mesosphere has an inverse relationship with the temperature
variations in the winter stratosphere. Further correlation analysis confirmed that the
dynamics in the winter stratosphere does have global influence on the atmospheric
mean state (*Karlsson et al.*, 2009; *Körnich and Becker*, 2010; *Tan et al.*, 2012). The
feedback between gravity-wave drag and zonal wind induced by mesospheric
cross-equatorial flow is a reasonable explanation for the inter-hemispheric coupling
mechanism. *Stray et al.* (2015) proposed that the enhancement of wave number 1 and
2 planetary waves at ~95 km could be a common feature during SSW period. Thus it
is reasonable to argue that the SSW may also have significant influence on QTDW
(*Lima et al.*, 2012).
A strong SSW event occurred in January 2006, when the QTDW activities also
exhibited abnormal behaviors consisting of an unusually strong W2 QTDW identified
in the wind and temperature fields besides the conventional W3 mode (*Varavut*
*Limpasuvan and Wu*, 2009). Meanwhile the W3 QTDW peaks with an extremely
short period of ~42 hours (*Gu et al.*, 2013a, b). It was suggested that these abnormal



QTDW activities may be related to the unusually strong summer easterly during the
same period. *McCormack et al.* (2009) proposed that the strong planetary waves
leading to the SSW event could influence the background zonal wind and the QTDW
forcing by enhancing the northward component of the residual circulation. This theory
was          supported          by          simulations          from          the          control
thermosphere-ionosphere-mesosphere-electrodynamics  general  circulation  model
(TIME-GCM), which show that the zonal mean zonal wind and the mean flow
instability become stronger during a SSW event (*Gu et al.*, 2016c). Besides, they also
reported the nonlinear interaction between W3 QTDW and the zonal wavenumber 1
stationary planetary wave (SPW1), which generates a W2 QTDW (*Gu et al.*, 2015).
Nevertheless, unrealistic QTDW and SPW1 forcing is utilized in their numerical
simulation to compensate strong dissipation at lower model boundary (~10 hPa),
which may result in artificial nonlinear coupling. Thus, the influence of SSW on
QTDW needs further investigation with more realistic atmospheric conditions.
In addition to ground-based and satellite observations, synoptic meteorological
datasets could be utilized to perform diagnostic analysis on the propagation and
amplification of QTDW. In this paper, the anomalous QTDW activities during the
major SSW period of January/February 2006 will be further investigated on the basis
of NOGAPS-ALPHA reanalysis dataset, which has been proven to be capable of
reproducing both SSW and QTDW activities under realistic atmospheric conditions
(*McCormack et al.*, 2009). This work sheds new light on the question whether or not
the SSW in the winter stratosphere has significant influence on the QTDW in the



summer mesosphere. The dataset and analysis are briefly described in section 2. Our
analysis results are presented in section 3, followed by a summary in section 4.**2.**
**Datasets and analysis**
**2.1 Aura/MLS temperature**

The Aura satellite was launched on July 15, 2004, which is a major component of

the NASA Earth Observing System (EOS). The Microwave Limb Sounder (MLS) is
one of the four instruments onboard the Aura satellite that measures emissions from
ozone, chlorine and other trace gases with a sun-synchronous orbit (covering two
local times at a given latitude from ~82°S-82°N) (Schwartz et al., 2008). Aura satellite
travels around the earth with a period of ~99 minutes, and thus the atmosphere is
sampled with ~14.5 circles per day. The version 3.3 Aura/MLS temperature dataset
ranges from 261 hPa to 0.001 hPa (~10-96 km) with a precision of 0.6 K in the lower
stratosphere and 2.5 K in the mesosphere. The highest vertical resolute of 3.6 km lies
at 31.6 hPa, which degrades to ~6 km at 0.01 hPa. A least squares fitting method is
utilized to extract the QTDW information in Aura/MLS temperature from December
2005   to   February   2006,   which   is   then   compared   with   the   results   from
NOGAPS-ALPHS reanalysis dataset.
**2.2 NOGAPS-ALPHA**

The  NOGAPS-ALPHA  reanalysis  model  is  developed  at  Naval  Research

Laboratory (NRL), which is the Advanced Level Physics High Altitude version of the
Navy  Operational  Global  Atmospheric  Prediction  System.  The  NRL  Atmospheric
Variational  Data  Assimilation  System  (NAVDAS)  is  adopted  to  incorporate  both
ground-based and satellite observations (*Daley and Barker*, 2001), including the





global temperature observations from Aura/MLS and TIMED/SABER instruments.
The observational datasets are updated every 6 hours through the NAVDAS.
Nevertheless, we use the hourly meteorological fields from NOGAPS-ALPHA to
study the QTDWs. Please refer to *Eckermann et al.* (2009) and *Siskind et al.* (2012)
for more information about the model and data assimilation.
The NOGAPS-ALPHA reanalysis datasets have been previously used to study
atmospheric tides and QTDWs. For example, *Lieberman et al.* (2015) studied the
short-term variability of the nonmigrating tide and its relationship with the nonlinear
interaction between stationary planetary wave and migrating tide. *Pancheva et al.*
(2016) analyzed the global distribution and seasonal variation of both eastward and
westward propagating QTDWs. In addition, the inter-annual variability of the
nonlinear interactions between QTDW and migrating diurnal tide has also been
investigated (*McCormack et al.*, 2010; *McCormack et al.*, 2014). Their analysis
results show that the NOGAPS-ALPHA reanalysis model is capable of capturing tidal
and planetary wave behaviors in the atmosphere. We will use a two-dimensional least
squares fitting to extract QTDW signals in the NOGAPS-ALPHA dataset.
**3. Results and Discussion**
**3.1 QTDWs in Aura/MLS temperature**
Figures 1a and 1c show the spectra of the Aura/MLS temperature observation at
~0.005 hPa during January 12-19 and 23-30 of 2006, when the W3 and W2 reach
maximum amplitudes (shown later by Figure 2). The MLS observations at ~40ºS and
~20ºS are utilized in Figures 1a and 1c, respectively. It is clear that the W3 and W2
QTDWs dominate the wave spectra with periods of ~42 and ~45 hours, respectively.



The vertical and global structures of the W3 and W2 are shown in Figures 1b and 1d.
Most of the W3 oscillations are limited to the southern hemisphere with maximum
amplitude of ~12 K at ~40ºN and 0.005 hPa. The temperature field of W2 exhibits
comparable perturbations in both hemispheres, though the branch in the southern
hemisphere is slightly stronger than that in the northern hemisphere. This is because
the larger phase speed of W2 results in more broadly distributed positive refractive
index, which enables its propagation in both hemispheres (*Liu et al.*, 2004; *Gu et al.*,
2016c). The temporal variations of the QTDWs in the summer mesosphere and the
zonal mean temperature in winter stratosphere are plotted in Figure 2. The W3 QTDW
grows as the development of SSW in early January, and reaches maximum amplitude
at around January 15. The W2 QTDW reaches maximum amplitude of ~6 K at around
January 27 with a minor peak of ~3 K at around January 10. Both the W2 and W3
QTDWs fade away after February 9, when the SSW also disappears and the
atmosphere returns to a climatological state. Figure 3 shows the comparison between
the QTDWs during 2005 and 2006. Abnormally strong W2 activities are observed
during January 2006, which are very weak during January 2005. Besides, the W3
QTDW is also stronger in January 2006. These QTDW activities agree well with the
results presented by *Limpasuvan and Wu* (2009) and *Tunbridge et al.* (2011). We will
then investigate whether the abnormal QTDW activities during January 2006 are
related to the major SSW event during the same episode with NOGAPS-ALPHA
reanalysis dataset.
**3.2 QTDWs in NOGAPS-ALPHA**





Figure 4 shows the analysis results of W2 and W3 from NOGAPS-ALPHA
during the same time period as Figure 1. The W3 and W2 QTDW signals are also
clearly indicated in the NOGAPS-ALPHA reanalysis datasets, and their vertical and
latitudinal temperature structures agree well with the results from Aura/MLS. Besides,
we found that the temporal variations of both W2 and W3 (Figure 5) are also
consistent with Aura/MLS observations (Figure 2). This is not strange since the
Aura/MLS and TIMED/SABER temperature datasets are major components
incorporated in the data assimilation at mesopause. We will also compare the wind
structures of QTDW from NOGAPS-ALPHA with those in previous literatures.
Figure 6 shows the zonal and meridional wind structures of W2 and W3 in
NOGAPS-ALPHA. The perturbations of W3 are nearly twice as strong as the W2.
Again, we can see that the latitudinal structures of W2 are more symmetric to the
equator than W3. The zonal and meridional winds of W3 peak in the southern
hemisphere with amplitudes of ~45 m/s and ~65 m/s at ~50ºS and ~40ºS, respectively.
The zonal wind of W2 peaks at ~20º-40º in both hemispheres with amplitudes of
~10-20 m/s, while the meridional wind of W2 maximizes at the equator with
amplitude of ~35-40 m/s. Generally, these results agree well with previous satellite
observations (*Limpasuvan and Wu*, 2009; *Gu et al.*, 2013a). Thus we conclude that
both the temperature and wind fields in NOGAPS-ALPHA are reasonable and
comparable with realistic atmospheric state, which can be utilized in the mechanical
study of the anomalous QTDW activities during January 2006.
It is proposed that the SSW may have significant influence on QTDW by





changing the mean flow (*Gu et al.*, 2016c). Thus we will first show how the
background wind influences the amplification of QTDWs. A necessary condition for
the occurrence of baroclinic/barotropic instability for zonal mean zonal wind is $\bar{q}_\varphi < 0$,
where $\bar{q}_\varphi$ is the latitudinal gradient of the quasi-geostrophic potential vorticity (*Liu et*
*al.*, 2004):
$$\bar{q}_\varphi = 2\Omega \cos\varphi - \left(\frac{(\bar{u}\cos\varphi)_\varphi}{a\cos\varphi}\right)_\varphi - \frac{a}{\rho}\left(\frac{f^2}{N^2}\rho\bar{u}_z\right)_z \qquad (1)$$
where $\bar{u}$, $a$, $\varphi$, $f$, $N$, $\Omega$, and $\rho$ are the zonal mean zonal wind, earth radius, latitude,
Coriolis parameter, Brunt-Väisällä frequency, angular speed of the earth's rotation,
and the background air density, z means the vertical gradient. Planetary waves can be
amplified by the instabilities through mean-flow interaction. It has been found that the
EP flux of QTDW grows dramatically after the over-reflection by its critical layer
near the unstable region (*Liu et al.*, 2004). The EP flux of planetary waves, (e.g.,
QTDW), can be calculated following *McCormack et al.* (2014):
$$\vec{F}_{EP} = \rho a \cos\varphi \begin{bmatrix} -\overline{v'u'} \\ \left[ f - \frac{(\bar{u}\cos\varphi)_\varphi}{a\cos\varphi} \right] \frac{\overline{v'\theta'}}{\bar{\theta}_z} \end{bmatrix} \qquad (2)$$
where *u'*, *v'*, and *θ'* are the zonal wind, meridional wind, and potential temperature
perturbations of planetary waves.

The barotropic/baroclinic instabilities of the mean flow and the EP flux of W2

and W3 are shown in Figure 7. It is clear that the W3 is more favorable to propagate
in the summer hemisphere, and is dramatically amplified by the mean flow
instabilities at middle latitude between 0.1 and 0.01 hPa. Nevertheless, the W2 is





capable of propagating in both hemispheres due to its broadly distributed refractive
index (*Gu et al.*, 2016c). The summer branch is also amplified by the instabilities
related to the easterly wind, while the winter branch propagates directly from the
lower atmosphere to mesosphere. *Liu et al.* (2004) has shown that the amplification of
QTDW through wave-mean flow interaction most easily occurs near its critical layer,
which is also indicated in our analysis.

Figure 2 has shown that both the QTDWs and the SSW peak in the middle and

late January, thus Figure 8 shows the comparison between the zonal mean zonal wind
during January 11-30 of 2005 and 2006. The zonal wind during the SSW period of
2006 shows two major differences compared with that in 2005. First, the westerly
wind in winter stratosphere reverses to easterly. The winter westerly reversal is one
key feature of major SSW, which is induced by the rapid growth of stationary
planetary waves and their momentum deposition to the background mean flow (*Liu*
*and Roble*, 2002). Second, the summer easterly wind in the mesosphere is enhanced.
The interhemispheric couplings during SSW period have been reported in previous
literatures (*Karlsson et al.*, 2007, 2009; *Körnich and Becker*, 2010). We then analyzed
the correlation between the temporal variations of the global zonal mean zonal wind
and the zonal mean temperature at 70ºN and 10 hPa, which increase dramatically
during a SSW event. The correlation coefficients are shown in Figure 9. The zonal
wind in the summer mesosphere at middle latitude shows a significant inverse
relationship with the temperature variations in the winter stratosphere. In the summer
hemisphere, the zonal mean zonal wind is westward in the upper stratosphere and





mesosphere; it will be enhanced when the temperature in winter stratosphere increases.
Thus, we conclude that the zonal wind anomaly during January 2006 is most likely
correlated with the SSW event.

We then show how these differences result in different QTDW behaviors during

2005 and 2006. The mean flow instabilities of the background wind and the critical
layers of W2 and W3 are shown in Figure 10. First the enhanced summer easterly in
the mesosphere results in stronger barotropic/baroclinic instability, which provides
larger forcing for the amplification of QTDW. This results in stronger W3 amplitude
during 2006 than that during 2005 (Figure 3). Besides, the stronger summer easterly
in the mesosphere also sustains a critical layer for W2 during 2006 at middle latitude,
which is not observed in 2005. The phase speed of planetary wave is inversely
proportional to both period and zonal wave numbers, thus the phase speed of W2 is
larger than W3. The existence of W2 critical layer nearby the instability region
facilitates the wave-mean flow interaction, through which the energy of mean flow is
transferred to W2 (*Liu et al.*, 2004). This results in abnormally strong W2 oscillations
in 2006 than that in 2005. *Gu et al.* (2013b) also noted that the W3 during 2006 peaks
with an extremely short period of ~42 hours (also shown by Figure 1 and 4), whereas
the period of W3 during austral summer tends to be longer (~52 hours) (*Palo et al.*,
2007; *Tunbridge et al.*, 2011; *Yue et al.*, 2012). The W3 QTDW with a longer period
has a slower phase speed. Figure 11 shows the comparison between the critical layers
of 42- and 52-hour W3 for the zonal mean state during 2006. The critical layer of the
42-hour W3 runs at the edge of the mean flow instability, which is totally surrounded



by the critical layer of the 52-hour W3. Thus the 52-hour QTDW signal has already
been reflected away by the critical layer before it reaches the unstable region and
cannot be amplified through wave-mean flow interaction (*Liu et al.*, 2004). Figure
10b also shows that both the critical layers of W3 and W2 run across the mean flow
instabilities in winter stratospheric region, whereas there is no significant positive EP
flux divergence near this region (Figure 12) as that shown in the summer mesosphere.
Positive EP flux divergence indicates the source for planetary waves. Thus we
conclude that the mean flow instability related to the winter westerly reversal during
SSW period is not as effective for the QTDW amplification as that in the summer
mesosphere.
**3.3 The nonlinear coupling between W3 and SPW1**
*Gu et al.* (2015) proposed that the nonlinear interaction between W3 and SPW1
could also provide sources for W2. We also calculated the nonlinear advection
between W3 and SPW1 following *Gu et al.* (2016c), which is shown Figure 13. The
nonlinear advection from TIME-GCM shows a significant peak at the lower boundary
(~10 hPa) in the winter stratosphere (Figure 13 of *Gu et al.* (2016c)), which is not
shown by our results from NOGAPS-ALPHA. Note that both the W3 and SPW1 is
forced at the lower model boundary in TIME-GCM (~10 hPa), which is much
stronger than realistic situation to compensate the large dissipation. Thus we conclude
that the nonlinear advection between W3 and SPW1 is in fact insignificant in the
winter stratosphere. Besides, the nonlinear advection also shows four peaks in the
mesosphere. The peak in polar winter mesosphere (~85ºN, 0.01 hPa) is most possibly



related to the strong wave number 1 component of the wind oscillations, which is
shown by Figure 14. Considering that the W2 is only favored to propagate at middle
and low latitudes (*Gu et al.*, 2016c), the nonlinear coupling between W3 and SPW1 in
the winter polar region maybe ineffective for the observed W2 perturbations. There
are both significant wind perturbations for W3 and SPW1 at low latitudes in the
northern hemisphere (Figure 14), and their nonlinear advection reaches ~12-15
m/s/day in this region. This agrees well with the result from TIME-GCM and possibly
contributes to the northern branch of W2 (Figure 7b). The EP flux divergence of W2
in Figure 12 also shows a source at ~10ºN between 0.01 and 0.001 hPa, which is
possibly related to the nonlinear advection between W3 and SPW1. The wind
perturbations of W3 reach maximum amplitude at middle and low latitudes in the
summer mesosphere, and the nonlinear advection also reaches ~15 m/s/day and ~9
m/s/day at ~50ºS and ~10ºS, respectively. These nonlinear couplings may contribute
to the southern branch of W2 (Figure 7b) and its positive EP flux divergence at ~25ºS
between 0.01 and 0.001 hPa (Figure 12a).
Though the W3 and SPW1 shows significant nonlinear coupling at middle and
low latitudes in the mesosphere, this does not mean that the nonlinear interaction is
the only source for W2. The EP flux of W2 in the winter stratosphere shows clear
upward propagation tendency, which most probably originates from the lower
atmosphere (Figure 15). The strong planetary wave activity in winter hemisphere,
which is responsible for the occurrence of SSW, may also provide strong sources for
QTDW in the lower atmosphere. *Gu et al.* (2016a, b) also showed that there are



persistent QTDW signals in the lower atmosphere, whereas the amplification of
QTDW in the mesosphere is dependent on the strength of the summer easterly. The
interhemispheric coupling during SSW period results in strong summer easterly jet in
January 2006, which provides suitable condition for the amplification of W2 signals
in the lower hemisphere.
**4. Summary**
In this paper, the influence of SSW on QTDWs is further investigated with
NOGAPS-ALPHA reanalysis dataset, which is a further contribution to previous work
reported by *Gu et al.* (2016c). Their TIME-GCM simulations use a climatological
atmosphere state as the background and the planetary waves are forced at the lower
model boundary (~10 hPa), which may induce artificial signals. Nevertheless, the
NOGAPS-ALPHA reanalysis dataset incorporates realistic observation from the
ground to mesosphere, which avoids the lower boundary effect. Our analysis shows
that the nonlinear interaction between W3 and SPW1 most probably occurs at middle
and low latitudes in the mesosphere.
During the major SSW period of January 2006, the QTDWs exhibit strong
oscillations with zonal wave number 2 and the conventional wave number 3 mode
peaks at an extremely short period. We found that the inter-hemispheric coupling
induced by strong winter planetary wave activities plays a crucial role in connecting
the winter stratospheric SSW and the summer mesospheric QTDW. To be exact, the
summer easterly is enhanced during a SSW event through the inter-hemispheric
coupling, which results in anomalous QTDW behaviors. The enhanced summer
easterly can sustain critical layers for QTDW with larger phase speed (e.g., smaller





zonal wave number, short period), which facilitate their amplification through
wave-mean flow interactions. Moreover, the enhanced summer easterly also provides
stronger barotropic/baroclinic instabilities and thus a larger forcing for the
amplification of QTDW, which results in strong W3 oscillation during January 2006.
Thus, we conclude that the abnormal QTDW activities in the summer mesosphere
observed by *Limpasuvan and Wu* (2009) are correlated with the major SSW event in
the winter stratosphere through inter-hemispheric coupling. We should note that the
summer easterly may also exhibits strong inter-annual variations, which could result
in different QTDW activities during other SSW years. A detailed comparison between
the QTDWs (with different zonal wave numbers) during SSW and non-SSW years
will be statistically studied in the future.
**Acknowledgement**
This work is sponsored by the Project Funded by China Postdoctoral Science
Foundation (2015M582001, 2016T90573), the National Natural Science Foundation
of China (41421063, 41304123), and Hundred Talents Program (D). The
NOGAPS-ALPHA dataset is available at ftp:map.nrl.navy.mil/pub/nrl/nogaps and the
Aura/MLS temperature observation can be downloaded by
https://disc.sci.gsfc.nasa.gov/Aura/data-holdings/MLS.



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

D05111.




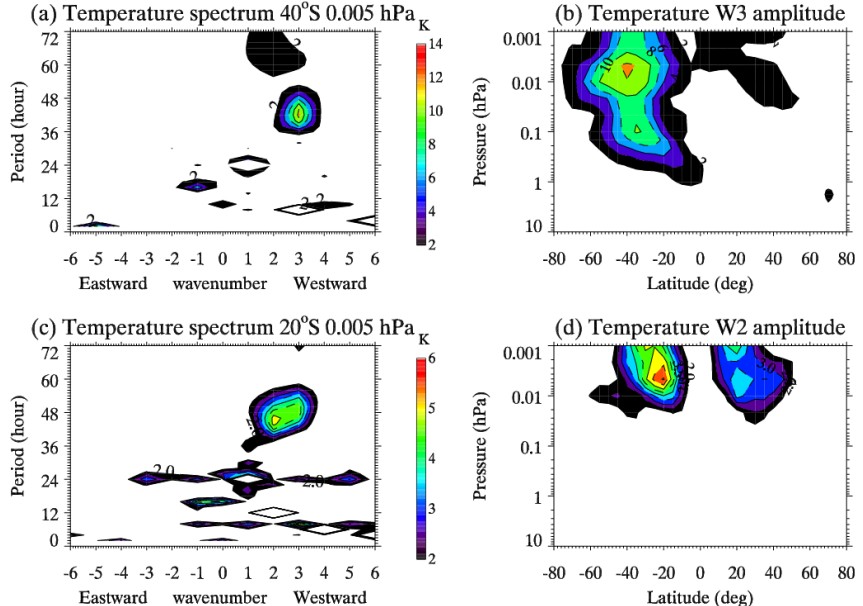


Figure 1 The wave number-period spectra of the Aura/MLS temperature observations
during (a) January 12-19 of 2006 at ~40°S and ~0.005 hPa, (c) January 23-30 of 2006
at ~20°S and ~0.005 hPa. The corresponding latitudinal and vertical structures of the
W3 and W2 QTDWs are shown in (b) and (d), respectively.





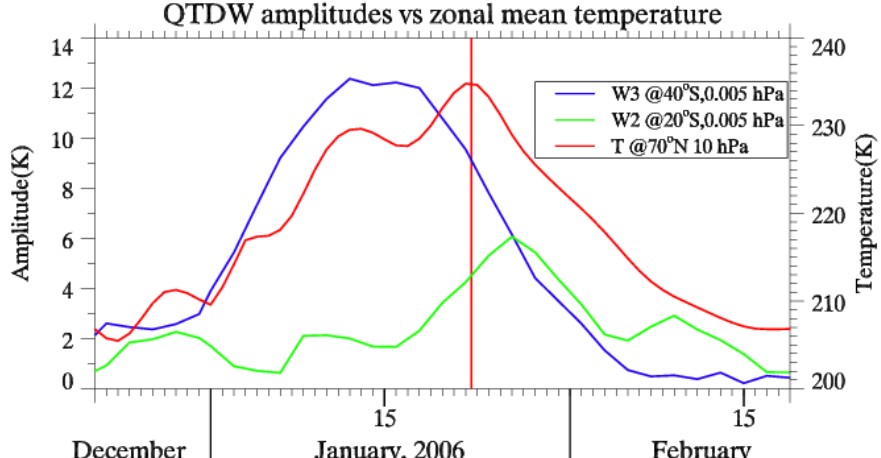

Figure 2 The temporal variations of the (blue) W3 at ~40°S and (green) W2 at ~20°S.
The zonal mean temperature at 70°N and 10 hPa is also plotted (red). The Aura/MLS
temperature observations are utilized in the analysis. The vertical red line indicates the
warming peak of the SSW.





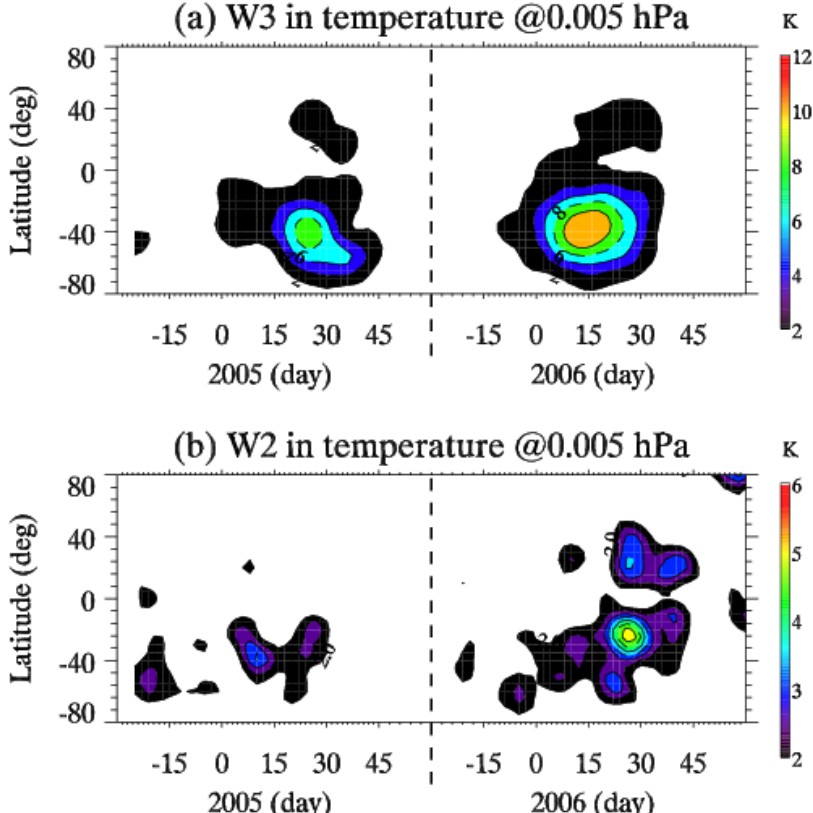


Figure 3 Temporal variations of the (a) W3 and (b) W2 in Aura/MLS temperature
observations at ~0.005 hPa during 2005 and 2006.





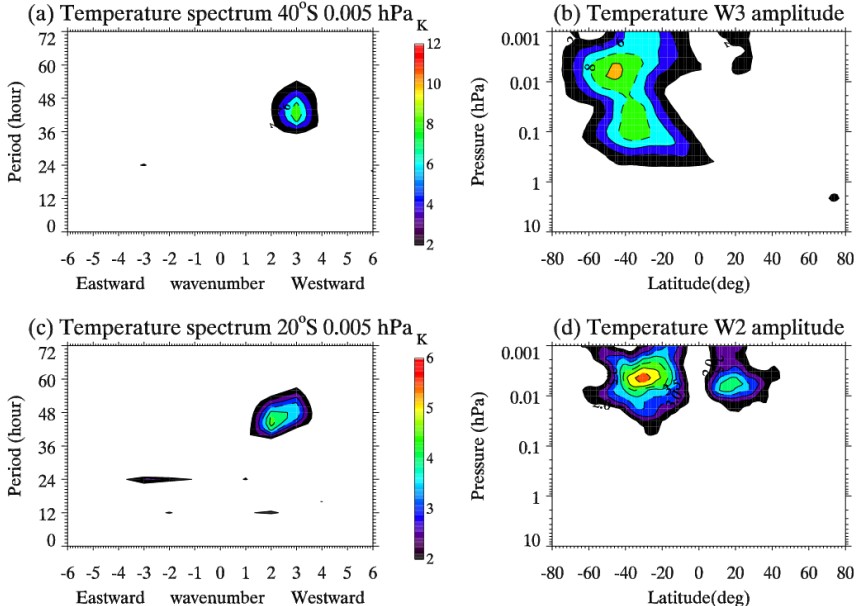


Figure 4 The same as Figure 1 but for the NOGAPS-ALPHA reanalysis datasets.






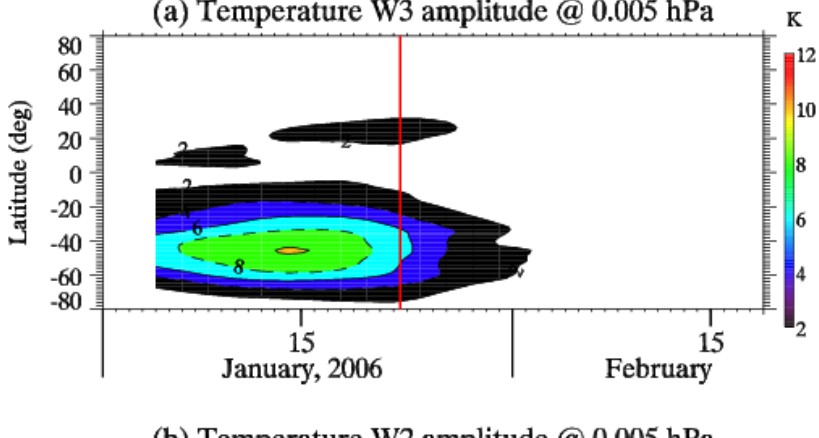

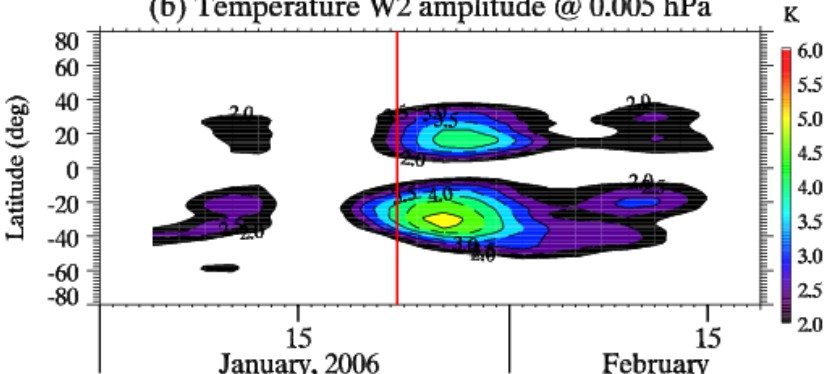


Figure 5 Temporal variations of the (a) W3 and (b) W2 QTDWs at ~0.005 hPa during
2006 from NOGAPS-ALPHA reanalysis dataset. The vertical red lines indicate the
warming peak of SSW.




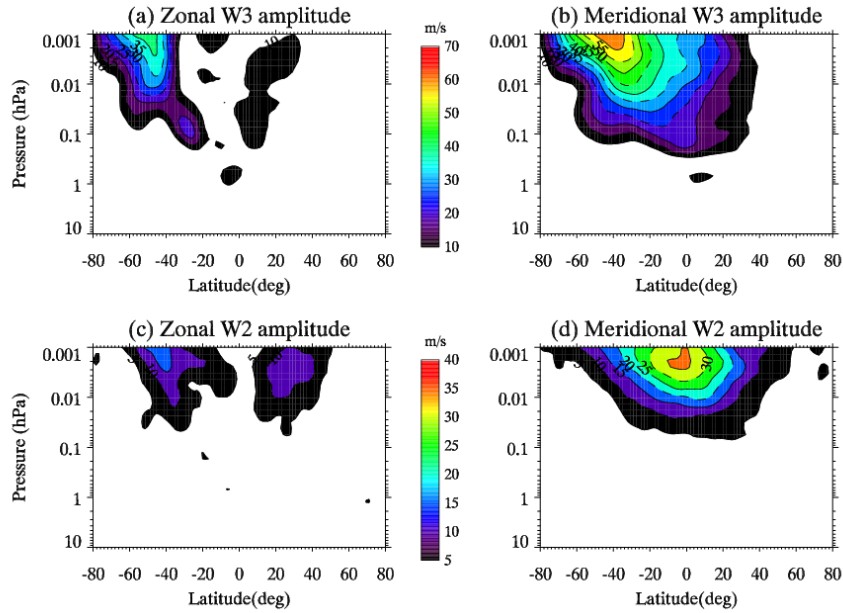


Figure 6 Altitude-latitude structures of the (a, b) W3 and (b, d) W2 in (a, c) zonal and
(b, d) meridional wind components. The wind fields during January 12-19 and 23-30
of 2006 are utilized for the analysis of W3 and W2, respectively.




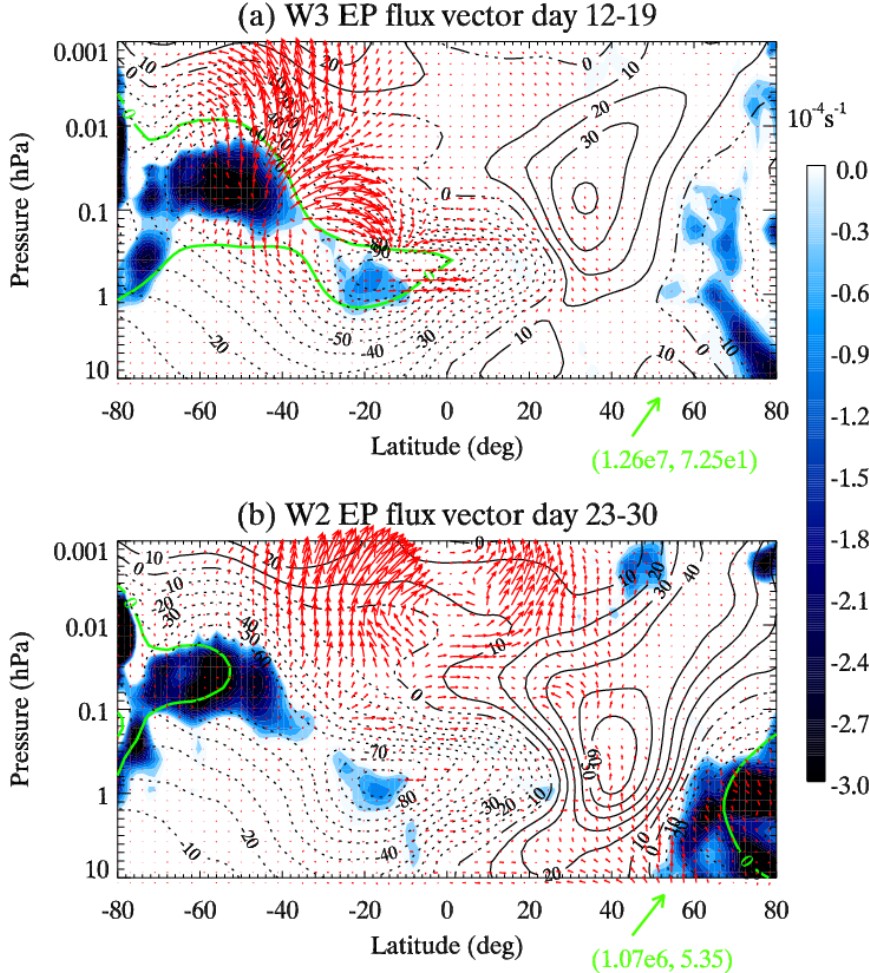


Figure 7 The EP flux vectors of (a) W3 during January 12-19 and (b) W2 during
January 23-30. The barotropically/baroclinically unstable regions ($\bar{q}_\varphi < 0$, equation 1)
are shaded with blue, and the critical layers are overplotted with green lines. The EP
flux vectors are normalized by the square root of the neutral density. The reference
lengths are shown at right bottom.



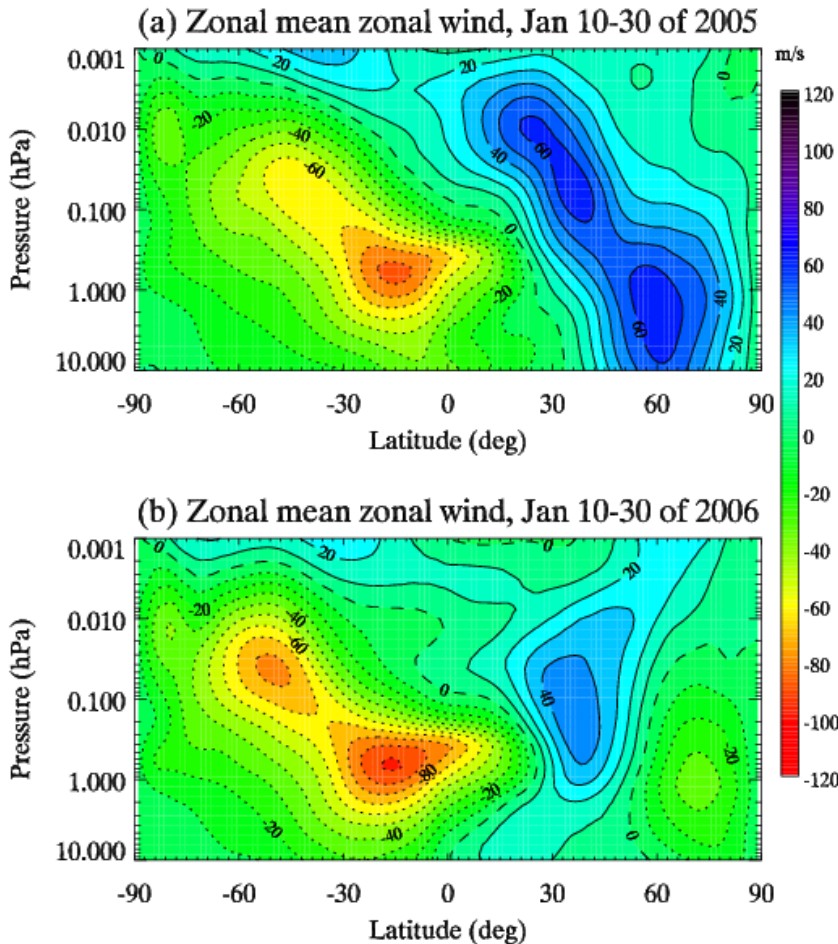


Figure 8 The zonal mean zonal wind during days 10-30 of (a) 2005 and (b) 2006. The
eastward and westward winds are plotted with solid and dotted lines, respectively.




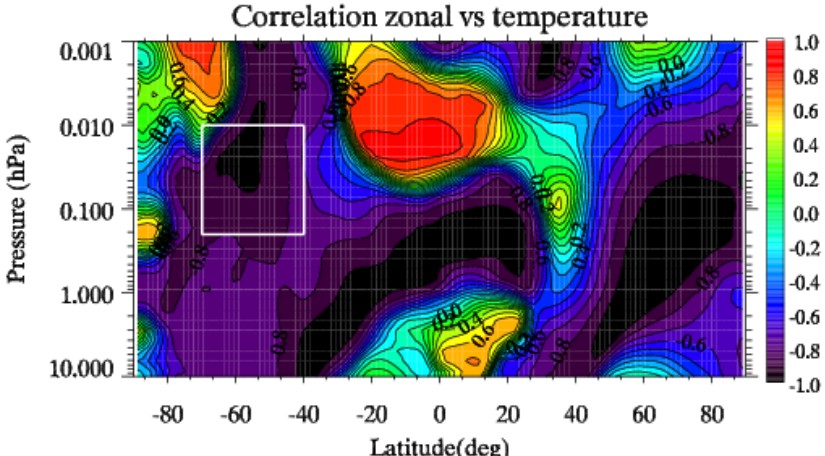


Figure 9 The correlation coefficient between the global zonal mean zonal wind and the temperature at 10 hPa and 70ºN from January 1 to February 20 of 2006. The rectangle indicates the unstable region that contributes most significantly to the amplification of QTDW.






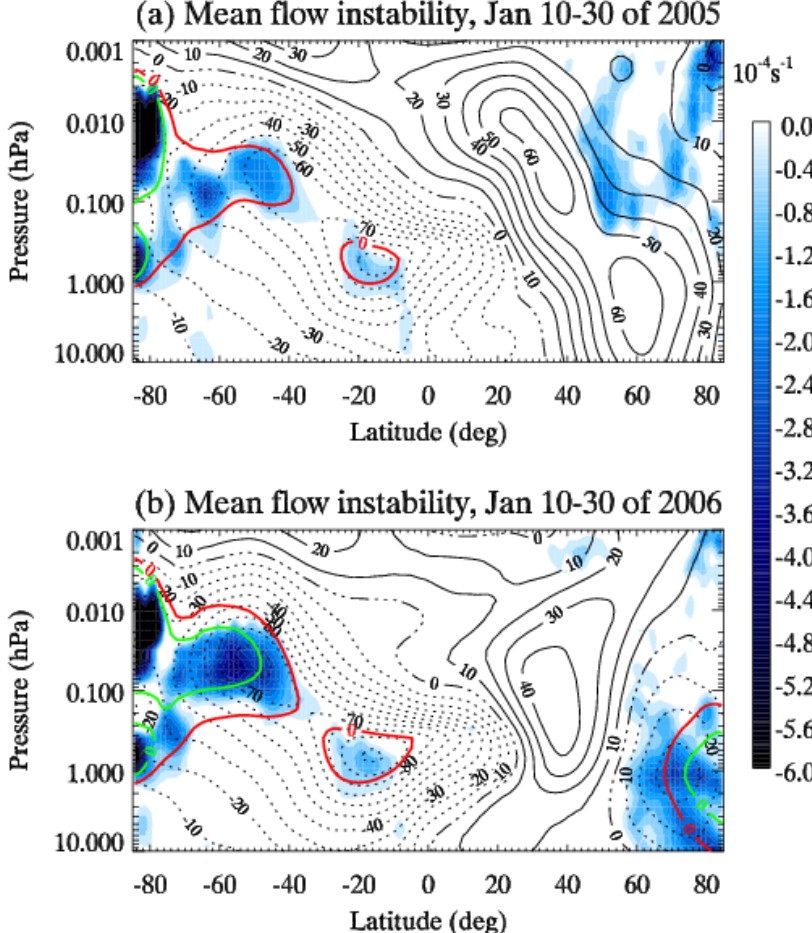


Figure 10 Comparison between the critical lines of the (red) 42-hour W3 and (light
green) 45-hour W2 for zonal mean zonal wind during days 10-30 of (a) 2005 and (b)
2006. The westward (eastward) zonal wind is plotted with dot (solid) lines, and the
barotropically/baroclinically unstable regions ($\bar{q}_\varphi < 0$, equation 1) are shaded with
blue.




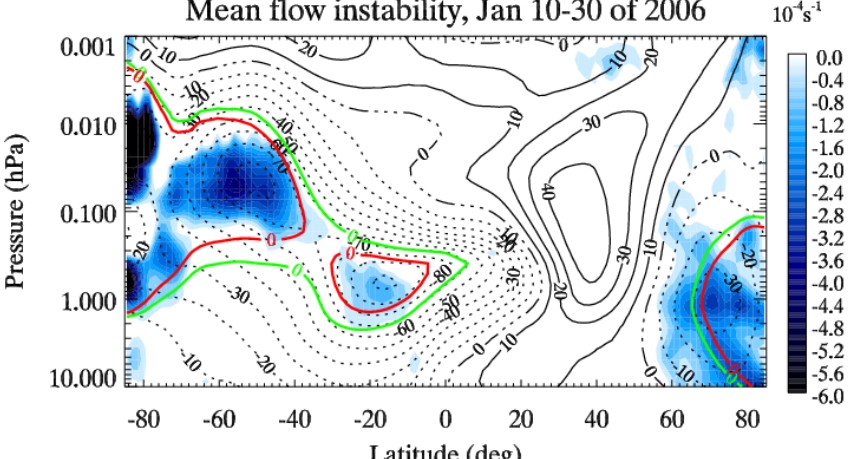


Figure 11 The same as Figure 10 but for the comparison between the critical lines of
the (red) 42-hour and (light green) 52-hour W3 QTDW during days 10-30 of 2006.



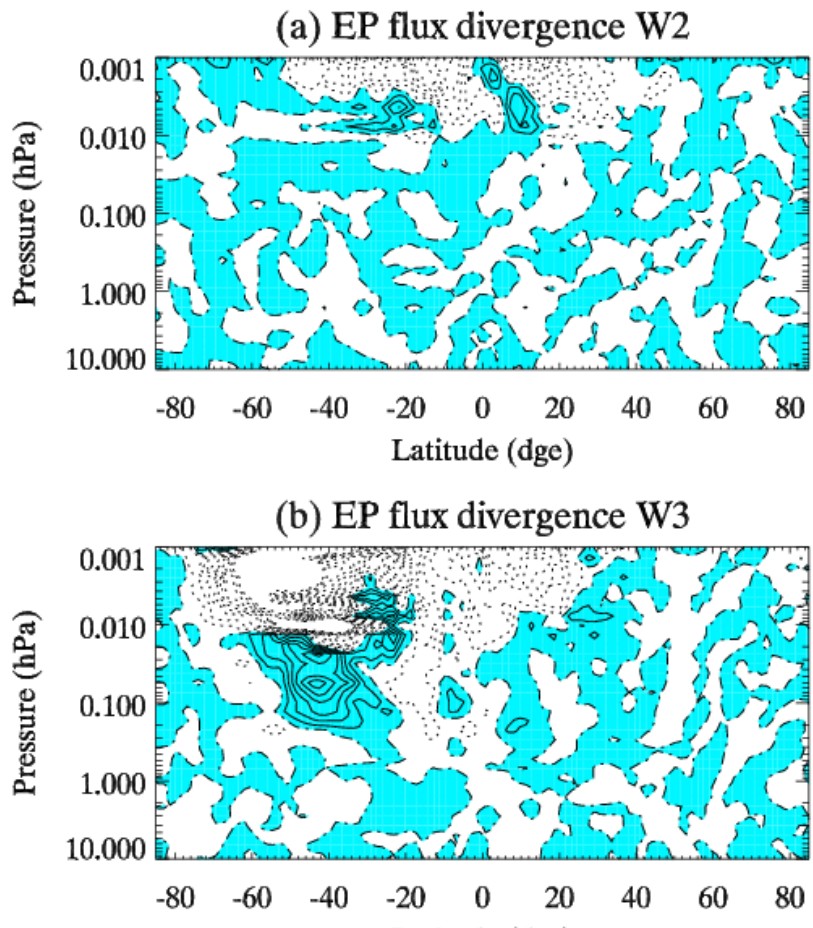


Figure 12 The EP flux divergence of (a) W2 and (b) W3 during January 23-30 of 2006.

The shaded region indicates positive EP flux divergence, and the contour interval is 2
m/s/day.



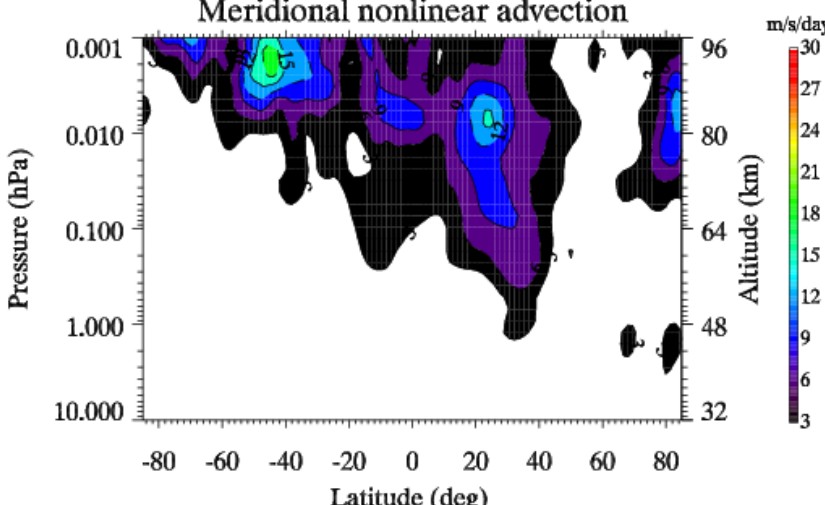


Figure 13 Meridional component of the nonlinear advection between W3 and SPW1
during January 23-30 of 2006.



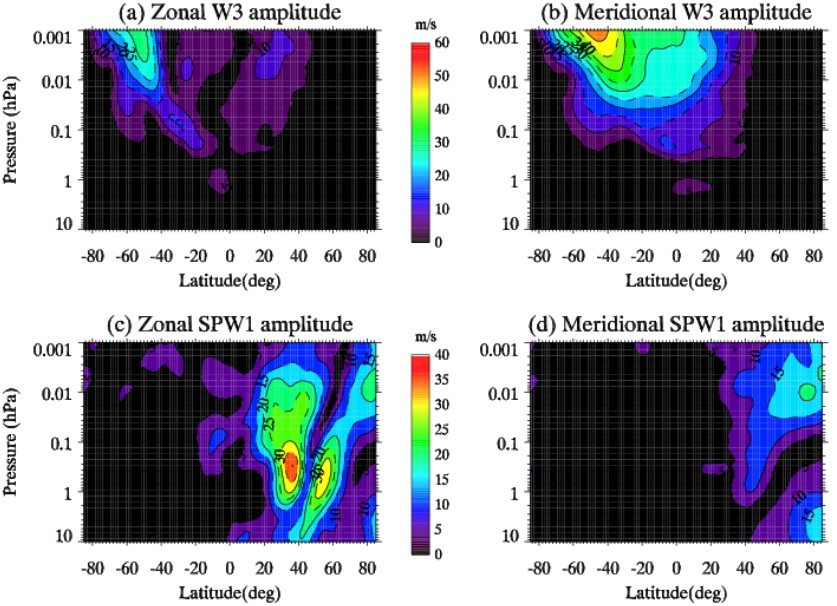


Figure 14. Altitude-latitude structures of (a, b) W3 and (c, d) SPW1 in (a, c) zonal and
(b, d) meridional winds during January 23-30 of 2006.


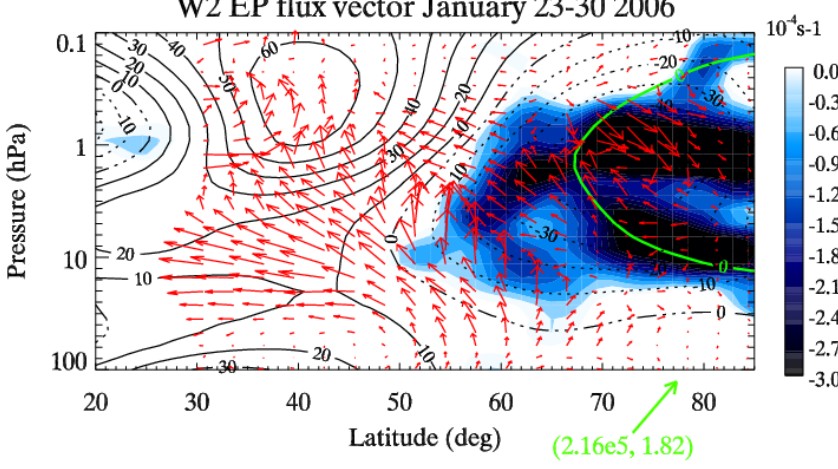


Figure 15. The EP flux vectors of W2 and the mean flow instabilities during January
23-30 near the winter stratosphere. The EP flux vectors are normalized by the square
root of the neutral density. The reference length is shown at right bottom.