# Peer review of "Manuscript under review for journal Atmos. Chem. Phys."

_Atmospheric Chemistry and Physics, 2017_

## Referee Comment (RC1) · Anonymous Referee #1 · 4 Aug 2017

Review of "Investigation on the abnormal quasi-two day wave activities during sudden stratospheric warming period of January 2006" by Gu et al. The paper investigated the impact of the SSW on QTDW at other hemisphere. The authors have carried out a number of analyses using satellite observations and reanalysis data to establish their conclusions. However, the findings do not illustrate any novelty in their characteristics. In this context it should be pointed out that earlier study by Gu et al. (2016c) already investigated on the SSW and QTDW (W2, W3) relationship using greater number of warming events including the present warming episode of 2006. Noting the listed issues the paper requires substantial revision before it could be considered for publication.

[Figure]

Major points 1) Characteristic features of the QTDW (W2 and W3) over both hemispheres during warming period as found from the present study are very similar with the past findings as expected. Explanation and interpretation related to the W2 and W3 components bear significant resemblance with the past study of Gu et al. (2016c).

2) The correlation analysis between zonal mean temperature (70N, 10 hPa) and zonal wind shown in Figure 9 looks very crude which does not entail any firm physical background, rather it is a chance occurrence. High anticorrelation is observed at several regions in the plot and cannot be considered as instability zones to excite QTDW. It is not clear why the interval is chosen from 1 Jan to 20 Feb when SSW activity is selected during 10-30 Jan for rest of the analyses.

3) The instability contours (Figures 7, 10, 11) do not suggest any evident link between the southern and northern hemispheres, rather southern hemispheric instabilities (to excite W3 and W2 QTDW near 40S and 20S, respectively) are found to be linked with highest instability southern polar mesosphere and stratosphere regions. Therefore from the present results it seems SSW has little to do with the summer hemisphere QTDW activity except coincidence and hence it requires substantiation.

4) Section 3.3: Fig. 14, Authors have mentioned that the present study could provide more realistic view of nonlinear interaction between SPW1 and W3 resulting generation of W2 QTDW as compared to the past study by Gu et al. (2016c). The present results (Figure 14) shows prominence of W3 in southern hemisphere and SPW1 in northern hemisphere indicating unlikely generation of W2 QTDW due to nonlinear interaction between SPW1 and W3 QTDW. Furthermore, nonlinear advection as shown in Figure 13 seems to be unreliable for identification of W2 generation as authors ruled out one such possibility in the text at winter polar region. Therefore using nonlinear advection to identify W2 QTDW generation looks incorrect in the present scenario.

5) Overall, the authors have pointed out stronger W2 and W3 QTDW in summer hemisphere in 2006 (SSW) in comparison with 2005 (no SSW), but it is not clear how the

disturbance propagates from winter to summer hemisphere. Therefore the authors are suggested to look into the latitudinal-temporal evolution of the dynamical parameters and attempt to link these two entities (SSW and QTDW in summer hemisphere) which could reduce present understanding gap and add significance to the present work since the past study by Gu et al. (2016c) already reported southern hemisphere QTDW enhancement through easterly wind strengthening during northern hemispheric SSW.

6) Figure 2, Instead of absolute temperature it is better to plot temperature deviation from mean wind.

7) Fig 7, 10, 11: Authors should describe how critical layers of W2 and W3 are calculated

8) Fig 12. It is not clear why EP flux divergence for W3 is plotted for the interval 23-30 Jan, whereas in all other cases it is 12-19 Jan for W3.

Other points 9) L29: Replace "provides stronger" by "strengthen"

10) L40: Replace "oscillation" by "significant variability"

11) L40: Replace "with a period. . .." By " with the period. . .."

12) L45-46: "with different wavenumbers. . .. . .". Please mention the wavenumbers

13) L-175: Correct "∼40N" to "40S"

14) L-214-215: Replace "in the mechanical study of the" by "in studying"

15) L-333: Correct "Their TIME-GCM simulations use" to "Their TIME-GCM simulations used"

16) L-355: Correct "may also exhibits" to "may also exhibit"

17) L-360-365: Both the links of data sources for NOGAPS-ALPHA and MLS-AURA are inaccessible. Please correct them.

---

## Referee Comment (RC2) · Anonymous Referee #2 · 15 Aug 2017

Referee report on "Investigation on the abnormal quasi-two day wave activities during sudden stratospheric warming period of January 2006"

This paper examines the unusual quasi two-day wave (QTDW) behavior during sudden stratospheric warming (SSW) period of January 2006, and reaches two main conclusions:

1. The unusually strong W2 QTDW is identified during the 2006 Austral summer, along with the conventional W3 component. 2. The strongest W2 signal occurs due to: (a) a manifestation of the summer easterly jet instability induced by SSW event via inter-hemispheric coupling and (b) a nonlinear interaction between W3 QTDW and wave

number 1 stationary planetary waves (SPW1).

Neither of these findings is new and the first is definitely not new. One can find a similar description in Limpasuvan and Wu (2009) and other previous studies by the same first author (e.g., Gu et al., 2016a,c). In particular, the unusual QTDW behavior during the 2006 Austral summer has been well documented in Limpasuvan and Wu (2009), where they showed that the conventionally dominant mode of QTDW with zonal wavenumber 3 (W3) is followed by a strong W2 component traveling westward (at nearly the same phase speed). In addition, the characteristic features of the QTDW (W2 and W3) found in this study are very similar to the previous findings of Gu et al. (2016c), who considered a large number of SSW events (including the warming episode of 2006). This also includes interpretation related to W2 generation by a nonlinear interaction between W3 and SPW1 during SSW events (Gu et al., 2016c). The authors will need to argue the significance of their work, with an emphasis on their novel findings. Because of these concerns, the manuscript requires a major and mandatory revision. If these concerns cannot be addressed, I would not recommend publishing this manuscript in ACP.

Other major points:

1. The interpretation of Fig. 7, regarding the source of W2 is not convincing. The EP flux vectors associated with W2 QTDW in the summer hemisphere are far from the instability source and the critical layers. Therefore, the argument for the amplification of QTDW via wave-mean flow interaction near the critical layer seems flimsy and requires further investigation. In addition, the QTDW activity in the winter branch seems to be partly originating from the tropical (0-20ËŽN) mesosphere. Can the authors explain this?

2. The authors also argued that the stronger W2 QTDW in austral summer 2006 is due to enhanced inter-hemisphere coupling induced by SSW in the winter hemisphere; however, such evidence is not clear from Fig. 9 and Fig. 10. I would suggest the authors prove it quantitatively, e.g., by calculating the residual mass-stream function

induced by resolved planetary wave drag (via downward control principle), as outlined by Lubis et al. (2016, Eq. 5), and the associated diabatic heating ($d\theta/dz$ w*). If the authors' argument is correct, hence, after the SSW event, we do expect an enhanced residual (inter-hemispheric) circulation in the summer hemisphere along with increased diabatic heating near the austral mesospheric jet.

3. Based on Figs 10-11, it is clear that the regions where the PV gradient is negative are potentially baroclinically or barotropically unstable, and thus represent potential sources of QTDW activity; however, it is still unclear which types of instability are more dominant for such processes; is it barotropic or baroclinic mode? Also, what causes the negative PV gradient in that region? Is this associated with changes in vertical shears or wind curvature? Please clarify.

4. The interpretation of Figs 13-14 is very confusing. The results shown in Figs 13 and 14 do not indicate that the W2 QTDW is generated via nonlinear advection interaction between W3 and SPW1. This is due to the fact that enhanced meridional nonlinear advection in the summer hemisphere (Fig. 13) is not accompanied by enhanced SPW1 activity (in u and v) in the same region (Fig. 14), rather only a prominence of W3 activity. Therefore, the generation of W2 QTDW via meridional nonlinear advection seems to be unlikely.

Specific points:

L47: Delete "to exist"

L57-58: Missing references

L59-L60: A similar finding was also reported by Lossow et al. (2015) and Lubis et al. (2016).

L87-L99: In addition to inter-hemispheric coupling induced by SSW events, a strengthening of summer mesospheric easterlies can also be induced by stratospheric ozone depletion in spring, leading to an increased instability of the summer mesospheric easterly jet and, thus, enhanced QTDW activity (see Lossow et al., 2015; Lubis et al., 2016).

L206-L207: Why is the latitudinal structure of W2 more symmetrical, compared to W3? Is this due to the characteristic of the instability-normal mode of the wave? Please explain.

L236-L237: Please clarify this result by plotting a latitude-height structure of the refractive index associated with W2?

L287: Why does positive EP flux divergence indicate the source of planetary waves?

L293: Please provide the nonlinear advection equation that you used in the manuscript.

Some references:

Limpasuvan, V., and D. L. Wu (2009), Anomalous two-day wave behavior during the 2006 austral summer, Geophys. Res. Lett., 36(4), L04807.

Lossow, S., C. McLandress, A. I. Jonsson, and T. G. Shepherd, 2012: Influence of the Antarctic ozone hole on the polar mesopause region as simulated by the Canadian Middle Atmosphere Model. J. Atmos. Sol.-Terr. Phys., 74, 111–123.

Lubis, S.W., N. Omrani, K. Matthes, and S. Wahl, 2016: Impact of the Antarctic Ozone Hole on the Vertical Coupling of the Stratosphere-Mesosphere-Lower Thermosphere System. J. Atmos. Sci., 73, 2509–2528.

---

## Author Comment (AC1) · 15 Sep 2017

We thank the reviewer for the constructive comments. Our responses are listed below. Interactive comment on "Investigation on the abnormal quasi-two day wave activities during sudden stratospheric warming period of January 2006" by Sheng-Yang Gu et al.

Anonymous Referee #1

Review of "Investigation on the abnormal quasi-two day wave activities during sudden stratospheric warming period of January 2006" by Gu et al. The paper investigated the

impact of the SSW on QTDW at other hemisphere. The authors have carried out a number of analyses using satellite observations and reanalysis data to establish their conclusions. However, the findings do not illustrate any novelty in their characteristics. In this context it should be pointed out that earlier study by Gu et al. (2016c) already investigated on the SSW and QTDW (W2, W3) relationship using greater number of warming events including the present warming episode of 2006. Noting the listed issues the paper requires substantial revision before it could be considered for publication.

Major points:

1) Characteristic features of the QTDW (W2 and W3) over both hemispheres during warming period as found from the present study are very similar with the past findings as expected. Explanation and interpretation related to the W2 and W3 components bear significant resemblance with the past study of Gu et al. (2016c).

Reply: Motivated by the observational work of Limpasuvan et al. (2009) entitled "anomalous two-day wave behavior during the 2006 austral summer", we would like to further investigate the reasons for the anomalous QTDW activities and whether they are related to the major SSW event during the same time. To perform diagnostic analysis on the propagation and amplification of QTDW, we need synoptic wind and temperature model datasets from the stratosphere to the mesosphere. As far as we know, the NOGAPS-ALPHA reanalysis dataset is the only reported dataset that is capable of capturing both SSW in the stratosphere and QTDWs in the mesosphere during January 2006 (McCormack et al., 2009), which can also be freely accessed on the website (ftp://map.nrl.navy.mil/pub/nrl/nogaps). Thus we choose to utilize the NOGAPS-ALPHA reanalysis dataset to further analyze the QTDWs during January 2006 with a focus to answer whether the anomalous QTDWs are related to the major SSW event during the same time.

In the previous ACP paper (Gu et al., 2016), we studied the influence of sudden stratospheric warming on quasi-two day wave with theoretical TIME-GCM numerical simulations. Our present results from NOGAPS-ALPHA reanalysis dataset show both similarities and differences.

The similarities include:

(1) the summer easterly is enhanced during the SSW period, which may provide stronger instabilities and thus lager forcing for the amplification of QTDW; (2) the nonlinear interaction between W3 QTDW and zonal wave number 1 stationary planetary wave (SPW1) could generate a W2 QTDW.

Our new findings and new work in this paper include:

(1) The TIME-GCM simulations showed that the W3 becomes weaker during SSW periods due to the nonlinear interaction and the energy transfer from parent wave (W3) to child wave (W2). Note that the forcing of W3 at the lower boundary of TIME-GCM is constant in the previous simulations. Nevertheless, in real atmosphere, the planetary wave signals in the winter stratosphere, where the source of QTDW exists, are very strong during a major SSW event. In this case, the QTDW forcing in the lower atmosphere may not be constraints for the QTDW oscillations in the upper atmosphere. On the contrary, the strong planetary wave activities during SSW periods would provide strong QTDW sources in the lower atmosphere and result in strong QTDW oscillations (including both W2 and W3) in the upper atmosphere, such as the January 2006 situation presented in this paper.

(2) The previous TIME-GCM simulations show that the W2 peaks earlier than W3 due to the larger phase speed and thus weak dissipation of W2. We should note that the W2 is generated immediately when the W3 and SPW1 are forced at the lower model boundary. In real atmosphere, the W2 QTDW may peak later than W3 according to the occurrence time of the nonlinear interaction between W3 and SPW1, such as the situation during January 2006.

(3) The TIME-GCM simulations show strong nonlinear advection between W3 QTDW and SPW1 at the lower model boundary ($\sim$10 hPa), which is weak in NOGAPS-ALPHA reanalysis datasets. Our present work made it clear that the strong nonlinear advection at $\sim$10 hPa in TIME-GCM is due to the lower boundary effect, which will possibly not occur in real atmosphere.

(4) According to previous statistical results (Gu et al., 2013), the period of QTDW is extremely short during 2006. Our analysis reveals that the short QTDW period is related to the enhanced summer easterly jet during the 2006 major SSW event. In other words, we found that the SSW can not only result in abnormally strong QTDW activities with different zonal wave numbers, but also influence the period of QTDW.

(5) Following the reviewers' comments, the barotropic and baroclinic instabilities are investigated separately. We found that the barotropic is $\sim$60-80% as strong as the baroclinic instability. In other words, the baroclinic instability would play a more important role in the amplification of QTDW, whereas the role of barotropic instability is also very important.

(6) In the revision, we also calculated the meridional and vertical circulations induced by the wave number 1 stationary planetary wave by using the downward control principle following Haynes et al. (1991) and Lubis et al. (2016). We found that the winter planetary wave induced circulations are confined to winter hemisphere, which would be ineffective for the inter-hemispheric coupling. This agrees well with the mechanism that the inter-hemispheric coupling is induced by the feedback between gravity wave breaking and zonal mean zonal wind in the mesosphere (Karlsson et al., 2009; Körnich and Becker, 2010). We also calculated the meridional component of the total residual circulation during and before the SSW event. Their difference shows that there is an anomalous cross-equator circulation from the winter to summer hemisphere in the mesosphere, which clearly suggests the inter-hemispheric coupling during SSW period.

(7) According to the analysis in this paper, we conclude that the abnormal QTDW activities during January 2006 are related to the major SSW event during the same time. The differences between the TIME-GCM and NOGAPS-ALPHA results are strengthened in the revision, and it is suggested that our present work with reanalysis dataset is a further contribution to the previous study with theoretical numerical simulation.

Limpasuvan, V., and D. L. Wu (2009), Anomalous two day wave behavior during the 2006 austral summer, Geophys. Res. Lett., 36, L04807.

McCormack, J. P., L. Coy, and K. W. Hoppel (2009), Evolution of the quasi 2-day wave during January 2006, J. Geophys. Res., 114, D20115.

Gu, S.-Y., T.Li, X. Dou, N.-N.Wang, D. Riggin, and D. Fritts (2013), Long-term observations of the quasi two-day wave by Hawaii MF radar, J. Geophys. Res. Space Physics, 118, 7886–7894, doi:10.1002/2013JA018858.

Gu, S. Y., H. L. Liu, X. Dou, and T. Li (2016), Influence of the sudden stratospheric warming on quasi-2-day waves, Atmos. Chem. Phys., 16(8), 4885-4896.

2) The correlation analysis between zonal mean temperature (70N, 10 hPa) and zonal wind shown in Figure 9 looks very crude which does not entail any firm physical background, rather it is a chance occurrence. High anticorrelation is observed at several regions in the plot and cannot be considered as instability zones to excite QTDW. It is not clear why the interval is chosen from 1 Jan to 20 Feb when SSW activity is selected during 10-30 Jan for rest of the analyses.

Reply: According to the definition of World Meteorological Organization, a SSW event is usually identified by the increase of temperature at 10 hPa (or below) poleward of 60N. Thus it is reasonable to choose a reference temperature point at 10 hPa and 70N to show the interhemispheric patterns during a SSW event. For example, Tan et al. [2012] also used a reference temperature point at 10 hPa and 76N-80N to study the interhemispheric couplings patterns during SSW periods. We focus on the zonal wind

due to the fact that the zonal wind is essential for the propagation and amplification of planetary waves.

We should note that a region with high anticorrelation does not mean instabilities there. The instability only occurs where the latitudinal gradient of the potential vorticity is negative. The diagnostics show that the mean flow instabilities during January 2006 exist in the summer easterly at middle and high latitudes, and in polar winter stratosphere due to the reversal of westerly. And the instabilities in the summer hemisphere are most effective for the amplification of QTDW. The high correlation patterns in other regions show that the SSW also has strong impact on the zonal wind there, but these variations are ineffective for the amplification of QTDWs.

Both the warming and QTDW peaks lie between Jan 10 and 30 of 2006, and we thus choose the dataset during Jan 10-30 when discussing the mechanism about QTDW amplification and its difference with that during 2005. We use a longer dataset (from Jan 1 to Feb 20) in the correlation analysis just to increase the reliability.

Tan, B., X. Chu, H.-L. Liu, C. Yamashita, and J. M. Russell III (2012), Zonal-mean global teleconnection from 15 to 110 km derived from SABER and WACCM, J. Geophys. Res., 117, D10106, doi:10.1029/2011JD016750.

3) The instability contours (Figures 7, 10, 11) do not suggest any evident link between the southern and northern hemispheres, rather southern hemispheric instabilities (to excite W3 and W2 QTDW near 40S and 20S, respectively) are found to be linked with highest instability southern polar mesosphere and stratosphere regions. Therefore from the present results it seems SSW has little to do with the summer hemisphere QTDW activity except coincidence and hence it requires substantiation.

Reply: We agree that Figures 7, 10 and 11 only show how the summer easterly and the corresponding instabilities influence the amplification of QTDW. In fact, the influence of the winter SSW on summer QTDW is realized just by the modulation on summer easterly through inter-hemispheric couplings. In other word, the QTDW is not triggered directly by the SSW, and the influence of SSW on QTDW is an indirect process. Our further correlation analysis in this paper, and the control TIME-GCM simulations in previous literature have confirmed the enhancement of summer easterly during the SSW period. In the revision, we also calculated the meridional component of the residual circulation, which shows an anomalous cross-equator circulation from the winter to summer mesosphere. This clearly suggests the inter-hemispheric coupling during SSW period.

4) Section 3.3: Fig. 14, Authors have mentioned that the present study could provide more realistic view of nonlinear interaction between SPW1 and W3 resulting generation of W2 QTDW as compared to the past study by Gu et al. (2016c). The present results (Figure 14) shows prominence of W3 in southern hemisphere and SPW1 in northern hemisphere indicating unlikely generation of W2 QTDW due to nonlinear interaction between SPW1 and W3 QTDW. Furthermore, nonlinear advection as shown in Figure 13 seems to be unreliable for identification of W2 generation as authors ruled out one such possibility in the text at winter polar region. Therefore using nonlinear advection to identify W2 QTDW generation looks incorrect in the present scenario.

Reply: The present analysis makes it clear that the strong nonlinear advection between W3 and SPW1 at ∼10 hPa, which is exhibited by previous TIME-GCM simulations, is due to the unrealistic wave forcing at the lower model boundary.

We note that the W3 peak mainly in the southern hemisphere and the SPW1 shows maximum wave amplitude in the northern hemisphere. We should also note that the zonal component of W3 has a weaker branch in the northern hemisphere with maximum amplitude at 20N, and the meridional wind perturbations of W3 can also propagate across the equator to as far as 40N in the northern hemisphere. Thus we suggest that the nonlinear advection between W3 and SPW1 most probably occur in the northern hemisphere equatorward of 40N. This generally agrees with the enhanced nonlinear advection at ∼20N in both previous TIME-GCM simulation and current NOGAPS-ALPHA reanalysis dataset.

Keep in mind that the nonlinear advection between W3 and SPW1 in fact represent the nonlinear effect on the mean flow tendencies. We use the nonlinear advection between W3 and SPW1 as a substitute to study their nonlinear interaction. We suggest that the nonlinear interaction most possibly occurs in the region with high nonlinear advection. We rule out the contribution of the nonlinear advection in polar winter region to the generation of W2 mainly due to that the W3 is extremely weak there and the W2 is not inclined to propagate in the polar region [Gu et al., 2016].

Gu, S. Y., H. L. Liu, X. Dou, and T. Li (2016), Influence of the sudden stratospheric warming on quasi-2-day waves, Atmos. Chem. Phys., 16(8), 4885-4896.

5) Overall, the authors have pointed out stronger W2 and W3 QTDW in summer hemisphere in 2006 (SSW) in comparison with 2005 (no SSW), but it is not clear how the disturbance propagates from winter to summer hemisphere. Therefore the authors are suggested to look into the latitudinal-temporal evolution of the dynamical parameters and attempt to link these two entities (SSW and QTDW in summer hemisphere) which could reduce present understanding gap and add significance to the present work since the past study by Gu et al. (2016c) already reported southern hemisphere QTDW enhancement through easterly wind strengthening during northern hemispheric SSW.

Reply: The SSW in the winter stratosphere induces inter-hemispheric couplings through the feedback between zonal mean state and gravity wave breaking in the mesosphere, which causes an abnormal mesospheric meridional circulation from the winter to summer hemisphere (Karlsson et al., 2009; Körnich, H., and E. Becker, 2010). The gravity wave drag is needed to illustrate the inter-hemispheric coupling processes in detail. It is a pity that the gravity wave drag is not included in the publicly accessed NOGAPS-ALPHA datasets. In the revision, we showed the difference between the meridional circulation during and before SSW, from which we can clearly see the cross-equator coupling from the winter to summer mesosphere during SSW period. The role of gravity wave breaking and the feedback between zonal mean state and the gravity

wave breaking needs our further investigation with other datasets.

Körnich, H., and E. Becker (2010), A simple model for the interhemispheric coupling of the middle atmosphere circulation, Advances in Space Research, 45(5), 661-668.

Karlsson, B., C. McLandress, and T. G. Shepherd (2009), Inter-hemispheric mesospheric coupling in a comprehensive middle atmosphere model, Journal of Atmospheric and Solar-Terrestrial Physics, 71(3–4), 518-530.

6) Figure 2, Instead of absolute temperature it is better to plot temperature deviation from mean wind.

Reply: Does the reviewer mean temperature deviation from mean temperature? In the revision, we showed the temperature deviation from seasonal (90-day) mean.

7) Fig 7, 10, 11: Authors should describe how critical layers of W2 and W3 are calculated

Reply: Added in the revision.

8) Fig 12. It is not clear why EP flux divergence for W3 is plotted for the interval 23-30 Jan, whereas in all other cases it is 12-19 Jan for W3.

Reply: Figure 12 is the previous manuscript is utilized to show whether instabilities related with the wind reversal during SSW period contribute significantly to the amplification of both W2 and W3. The wind reversal is much stronger during day 23-30 than during day 12-19, and thus we showed the EP flux divergence of both W2 and W3 during day 23-30.

Other points 9) L29: Replace "provides stronger" by "strengthen"

Reply: Done.

10) L40: Replace "oscillation" by "significant variability"

Reply: Done.

11) L40: Replace "with a period: : :." By " with the period: : :."

Reply: Done.

12) L45-46: "with different wavenumbers: : :: : :". Please mention the wavenumbers

Reply: Added in the revision.

13) L-175: Correct "_40N" to "40S"

Reply: Corrected.

14) L-214-215: Replace "in the mechanical study of the" by "in studying"

Reply: Done.

15) L-333: Correct "Their TIME-GCM simulations use" to "Their TIME-GCM simulations used"

Reply: Corrected.

16) L-355: Correct "may also exhibits" to "may also exhibit"

Reply: Corrected.

17) L-360-365: Both the links of data sources for NOGAPS-ALPHA and MLS-AURA are inaccessible. Please correct them.

Reply: The links are updated in the revision.

Please also note the supplement to this comment:
https://www.atmos-chem-phys-discuss.net/acp-2017-563/acp-2017-563-AC1-supplement.pdf

---

## Author Comment (AC2) · 15 Sep 2017

We thank the reviewer for the constructive comments. Our responses are listed below.
Referee report on "Investigation on the abnormal quasi-two day wave activities during sudden stratospheric warming period of January 2006"

[Figure]

This paper examines the unusual quasi two-day wave (QTDW) behavior during sudden stratospheric warming (SSW) period of January 2006, and reaches two main conclusions:

1. The unusually strong W2 QTDW is identified during the 2006 Austral summer, along with the conventional W3 component. 2. The strongest W2 signal occurs due to: (a) a manifestation of the summer easterly jet instability induced by SSW event via interhemispheric coupling and (b) a nonlinear interaction between W3 QTDW and wave number 1 stationary planetary waves (SPW1).

Neither of these findings is new and the first is definitely not new. One can find a similar description in Limpasuvan and Wu (2009) and other previous studies by the same first author (e.g., Gu et al., 2016a,c). In particular, the unusual QTDW behavior during the 2006 Austral summer has been well documented in Limpasuvan and Wu (2009), where they showed that the conventionally dominant mode of QTDW with zonal wavenumber 3 (W3) is followed by a strong W2 component traveling westward (at nearly the same phase speed). In addition, the characteristic features of the QTDW (W2 and W3) found in this study are very similar to the previous findings of Gu et al. (2016c), who considered a large number of SSW events (including the warming episode of 2006). This also includes interpretation related to W2 generation by a nonlinear interaction between W3 and SPW1 during SSW events (Gu et al., 2016c). The authors will need to argue the significance of their work, with an emphasis on their novel findings. Because of these concerns, the manuscript requires a major and mandatory revision. If these concerns cannot be addressed, I would not recommend publishing this manuscript in ACP.

Reply: Our present work is in fact motivated by the observational work of Limpasuvan et al. (2009) entitled "anomalous two-day wave behavior during the 2006 austral summer". We would like to further investigate the reasons for the anomalous QTDW activities and whether they are related to the major SSW event during the same time. To perform diagnostic analysis on the propagation and amplification of QTDW, we need synoptic wind and temperature model datasets from the stratosphere to the meso-

sphere. As far as we know, the NOGAPS-ALPHA reanalysis dataset is the only reported dataset that is capable of capturing both SSW in the stratosphere and QTDWs in the mesosphere during January 2006 (McCormack et al., 2009), which can also be freely accessed on the website (ftp://map.nrl.navy.mil/pub/nrl/nogaps). Thus we choose to utilize the NOGAPS-ALPHA reanalysis dataset to further analyze the QTDWs during January 2006 with a focus to answer whether the anomalous QTDWs are related to the major SSW event during the same time.

In the previous ACP paper (Gu et al., 2016), we studied the influence of sudden stratospheric warming on quasi-two day wave with theoretical TIME-GCM numerical simulations. Our present results from NOGAPS-ALPHA reanalysis dataset show both similarities and differences.

The similarities include:

(1) the summer easterly is enhanced during the SSW period, which may provide stronger instabilities and thus lager forcing for the amplification of QTDW; (2) the nonlinear interaction between W3 QTDW and zonal wave number 1 stationary planetary wave (SPW1) could generate a W2 QTDW.

Our new findings and new work in this paper include:

(1) The TIME-GCM simulations showed that the W3 becomes weaker during SSW periods due to the nonlinear interaction and the energy transfer from parent wave (W3) to child wave (W2). Note that the forcing of W3 at the lower boundary of TIME-GCM is constant in the previous simulations. Nevertheless, in real atmosphere, the planetary wave signals in the winter stratosphere, where the source of QTDW exists, are very strong during a major SSW event. In this case, the QTDW forcing in the lower atmosphere may not be constraints for the QTDW oscillations in the upper atmosphere. On the contrary, the strong planetary wave activities during SSW periods would provide strong QTDW sources in the lower atmosphere and result in strong QTDW oscillations (including both W2 and W3) in the upper atmosphere, such as the January 2006 situation presented in this paper.

(2) The previous TIME-GCM simulations show that the W2 peaks earlier than W3 due to the larger phase speed and thus weak dissipation of W2. We should note that the W2 is generated immediately when the W3 and SPW1 are forced at the lower model boundary. In real atmosphere, the W2 QTDW may peak later than W3 according to the occurrence time of the nonlinear interaction between W3 and SPW1, such as the situation during January 2006.

(3) The TIME-GCM simulations show strong nonlinear advection between W3 QTDW and SPW1 at the lower model boundary ($\sim$10 hPa), which is weak in NOGAPS-ALPHA reanalysis datasets. Our present work made it clear that the strong nonlinear advection at $\sim$10 hPa in TIME-GCM is due to the lower boundary effect, which will possibly not occur in real atmosphere.

(4) According to previous statistical results (Gu et al., 2013), the period of QTDW is extremely short during 2006. Our analysis reveals that the short QTDW period is related to the enhanced summer easterly jet during the 2006 major SSW event. In other words, we found that the SSW can not only result in abnormally strong QTDW activities with different zonal wave numbers, but also influence the period of QTDW.

(5) Following the reviewers' comments, the barotropic and baroclinic instabilities are investigated separately. We found that the barotropic is $\sim$60-80% as strong as the baroclinic instability. In other words, the baroclinic instability would play a more important role in the amplification of QTDW, whereas the role of barotropic instability is also very important.

(6) In the revision, we also calculated the meridional and vertical circulations induced by the wave number 1 stationary planetary wave by using the downward control principle following Haynes et al. (1991) and Lubis et al. (2016). We found that the winter planetary wave induced circulations are confined to winter hemisphere, which would be ineffective for the inter-hemispheric coupling. This agrees well with the mechanism that the inter-hemispheric coupling is induced by the feedback between gravity wave breaking and zonal mean zonal wind in the mesosphere (Karlsson et al., 2009; Körnich and Becker, 2010). We also calculated the meridional component of the total residual circulation during and before the SSW event. Their difference shows that there is an anomalous cross-equator circulation from the winter to summer hemisphere in the mesosphere, which clearly suggests the inter-hemispheric coupling during SSW period.

(7) According to the analysis in this paper, we conclude that the abnormal QTDW activities during January 2006 are related to the major SSW event during the same time. The differences between the TIME-GCM and NOGAPS-ALPHA results are strengthened in the revision, and it is suggested that our present work with reanalysis dataset is a further contribution to the previous study with theoretical numerical simulation.

Limpasuvan, V., and D. L. Wu (2009), Anomalous two day wave behavior during the 2006 austral summer, Geophys. Res. Lett., 36, L04807.

McCormack, J. P., L. Coy, and K. W. Hoppel (2009), Evolution of the quasi 2-day wave during January 2006, J. Geophys. Res., 114, D20115.

Gu, S.-Y., T.Li, X. Dou, N.-N.Wang, D. Riggin, and D. Fritts (2013), Long-term observations of the quasi two-day wave by Hawaii MF radar, J. Geophys. Res. Space Physics, 118, 7886–7894, doi:10.1002/2013JA018858.

Gu, S. Y., H. L. Liu, X. Dou, and T. Li (2016), Influence of the sudden stratospheric warming on quasi-2-day waves, Atmos. Chem. Phys., 16(8), 4885-4896.

Other major points:

1. The interpretation of Fig. 7, regarding the source of W2 is not convincing. The EP flux vectors associated with W2 QTDW in the summer hemisphere are far from the instability source and the critical layers. Therefore, the argument for the amplification of QTDW via wave-mean flow interaction near the critical layer seems flimsy and requires

further investigation. In addition, the QTDW activity in the winter branch seems to be partly originating from the tropical (0-20N) mesosphere. Can the authors explain this?

Reply: We agree that there are differences between the W2 and W3 QTDWs. For one thing, the W3 QTDW is more obviously amplified by the mean instabilities, while the W2 QTDW looks more like a free traveling planetary wave, whose amplitude grows with the decrease of atmospheric density. For another, the W3 QTDW is only favored to propagate in the southern (summer) hemisphere, while the W2 QTDW is capable of propagating in both hemispheres. This is partly due to the larger phase speed of W2, which results in weaker dissipation when propagating upward. In fact, there are some (though weak) clues at 20-40S and 0.1-0.01 hPa showing the W2 EP flux propagating away from the instability region, which indicates weak amplification through wave-mean interaction. There are two possible sources for the winter branch of W2: One, it is generated by the nonlinear interaction between W3 and SPW1; Second, it may originate from the source region in the lower atmosphere, and then directly propagates upward into the mesosphere. The differences between W2 and W3 are expressed more clearly in the revision.

2. The authors also argued that the stronger W2 QTDW in austral summer 2006 is due to enhanced inter-hemisphere coupling induced by SSW in the winter hemisphere; however, such evidence is not clear from Fig. 9 and Fig. 10. I would suggest the authors prove it quantitatively, e.g., by calculating the residual mass-stream function induced by resolved planetary wave drag (via downward control principle), as outlined by Lubis et al. (2016, Eq. 5), and the associated diabatic heating ($d\theta/dz$ w*). If the authors' argument is correct, hence, after the SSW event, we do expect an enhanced residual (inter-hemispheric) circulation in the summer hemisphere along with increased diabatic heating near the austral mesospheric jet.

Reply: The W2 QTDW has a larger phase speed compared with other QTDWs, which needs a stronger summer easterly to sustain a critical layer of W2. The inter-hemispheric couplings during SSW period result in an enhanced summer easterly,

which facilitates the existence of W2 critical layer and the wave-mean interaction.

We calculated the stream function induced by wave number 1 planetary wave, and the corresponding vertical and meridional residual circulations. The results show that the winter planetary wave induced circulations are confined in the winter hemisphere. This agrees well with the inter-hemispheric mechanism that the inter-hemispheric coupling arises from a feedback between the gravity wave breaking and the zonal wind in the mesosphere. We thus should calculate the stream function induced by gravity wave drag to illustrate the inter-hemispheric couplings. It is a pity that the gravity wave drag is not included in the publicly accessed NOGAPS-ALPHA dataset. In the revision, we showed the difference between the total meridional circulations during and before SSW. It is clear that there is an anomalous cross-equator mesospheric meridional circulation from the northern (winter) to southern hemisphere during SSW period, which provides evidence for the inter-hemispheric coupling.

3. Based on Figs 10-11, it is clear that the regions where the PV gradient is negative are potentially baroclinically or barotropically unstable, and thus represent potential sources of QTDW activity; however, it is still unclear which types of instability are more dominant for such processes; is it barotropic or baroclinic mode? Also, what causes the negative PV gradient in that region? Is this associated with changes in vertical shears or wind curvature? Please clarify.

Reply: In the revision, we added a Figure to show the barotropic and baroclinic instability separately. We found that the barotropic instability is usually ∼60-80% as strong as the baroclinic instability at middle latitudes in the southern mesosphere, where it is more effective for the amplification of QTDW. In other words, the wind vertical shears contribute more to the growth of QTDW, but the wind curvatures are also very important.

4. The interpretation of Figs 13-14 is very confusing. The results shown in Figs 13 and 14 do not indicate that the W2 QTDW is generated via nonlinear advection interaction

between W3 and SPW1. This is due to the fact that enhanced meridional nonlinear advection in the summer hemisphere (Fig. 13) is not accompanied by enhanced SPW1 activity (in u and v) in the same region (Fig. 14), rather only a prominence of W3 activity. Therefore, the generation of W2 QTDW via meridional nonlinear advection seems to be unlikely.

Reply: In fact, there is a minor peak of 6-9 m/s for the zonal component of SPW1 at 40S nearby the enhanced nonlinear advection. The enhanced nonlinear advection between W3 and SPW1 in the summer hemisphere corresponds to the nearby major peak of W3 and this minor peak of SPW1. We use the nonlinear advection between W3 and SPW1 as a substitute to study their nonlinear interaction. Our point is that the nonlinear interaction between W3 and SPW1 is more likely to occur in the region with strong nonlinear advection, whereas we are not sure whether there is a one to one correspondence between each other. For example, the enhanced nonlinear advection in winter polar mesosphere is probably not a source for W2, since the W2 is more favored to propagate at middle and low latitudes. We thus suggest that the enhanced nonlinear advections between W3 and SPW1 at 40-50S and 0-10S are possible sources for W2, while the enhanced nonlinear advection at 30N is a more conceivable source for W2.

Specific points:

L47: Delete "to exist"

Reply: Deleted in the revision.

L57-58: Missing references

Reply: Added in the revision.

L59-L60: A similar finding was also reported by Lossow et al. (2015) and Lubis et al. (2016).

Reply: These references are added in the revision.

L87-L99: In addition to inter-hemispheric coupling induced by SSW events, a strengthening of summer mesospheric easterlies can also be induced by stratospheric ozone depletion in spring, leading to an increased instability of the summer mesospheric easterly jet and, thus, enhanced QTDW activity (see Lossow et al., 2015; Lubis et al., 2016).

Reply: We agree that the ozone deletion in the southern hemisphere stratosphere will result in stronger easterlies in the mesosphere, which may also contribute to the enhancement of planetary wave activities. Nevertheless, this does not influence our conclusion that the enhanced summer easterly and QTDW activities during January 2006 are most likely related to the SSW event. For one thing, the O3 depletion occurs mainly during southern spring (Sep-Nov), whereas the QTDW is a summer phenomenon (Dec-Feb). Thus the enhanced instabilities induced by the ozone depletion may be ineffective for the amplification of QTDW; For another, the O3 mixing ratios during January 2005 and 2006 from Aura/MLS do not show significant differences that may account for the enhanced summer mesospheric easterly in 2006. Both the two references are cited and discussed in the revision.

L206-L207: Why is the latitudinal structure of W2 more symmetrical, compared to W3? Is this due to the characteristic of the instability-normal mode of the wave? Please explain.

Reply: The phase speed of W2 is larger, and therefore its waveguide has a broader latitudinal extension, which enables the propagation of W2 in both hemispheres. The larger phase speed also makes W2 less vulnerable to dissipation and critical layer filtering by the background wind when propagating upward (Gu et al., 2016).

Gu, S. Y., H. L. Liu, X. Dou, and T. Li (2016), Influence of the sudden stratospheric warming on quasi-2-day waves, Atmos. Chem. Phys., 16(8), 4885-4896.

L236-L237: Please clarify this result by plotting a latitude-height structure of the refractive index associated with W2?
Reply: A new Figure with refractive index of both W2 and W3 is added in the revision.

L287: Why does positive EP flux divergence indicate the source of planetary waves?

Reply: The 'source' here means energy conversion to planetary waves due to mean flow interactions. When the planetary wave is amplified by the mean flow instabilities, the EP flux divergence usually shows a positive value near the instability region. It is expressed more clearly in the revision.

L293: Please provide the nonlinear advection equation that you used in the manuscript.

Reply: Added in the revision.

Please also note the supplement to this comment:
https://www.atmos-chem-phys-discuss.net/acp-2017-563/acp-2017-563-AC2-
supplement.pdf

[Figure]

**Supplement:**

[revised manuscript text omitted]
has been found that the EP flux of QTDW grows dramatically after the over-reflection
by its critical layer (where the zonal mean zonal wind equals to the planetary wave
speed) near the unstable region (*Liu et al.*, 2004). The EP flux of planetary waves,
(e.g., QTDW), can be calculated following *McCormack et al.* (2014):

245
$$\vec{F}_{EP} = \rho a \cos \varphi \left[ f - \frac{\vec{v} \cdot \vec{u}}{a \cos \varphi} \frac{\vec{v} \cdot \vec{\theta}}{\vec{\theta}_z} \right]$$
 (2)

where *u*', *v*', and  $\theta$ ' are the zonal wind, meridional wind, and potential temperature perturbations of planetary waves. The phase speed of planetary wave can be calculated by  $(2\pi \cdot a)/(s \cdot T)$ , where the *s* and T are the zonal wave number and period, respectively.

The barotropic/baroclinic instabilities of the mean flow and the EP flux of W2 250 and W3 are shown in Figure 7. It is clear that the W3 is more favorable to propagate 251 in the summer hemisphere, and is dramatically amplified by the mean flow 252 253 instabilities at middle latitude between 0.1 and 0.01 hPa. Nevertheless, the W2 is capable of propagating in both hemispheres due to its more broadly distributed 254 refractive index (Gu et al., 2016c), which is also shown by Figure 8. The summer 255 branch is also amplified by the instabilities related to the easterly wind, while the 256 winter branch propagates directly from the lower atmosphere to mesosphere. Liu et al. 257 (2004) has shown that the amplification of QTDW through wave-mean flow 258 interaction most easily occurs near its critical layer, which is also indicated in our 259 analysis. Compared with W3 QTDW, which is more obviously amplified by the mean 260

| 261 | instabilities, the W2 QTDW looks more like a free traveling planetary wave. There        |
|-----|------------------------------------------------------------------------------------------|
| 262 | are only very weak clues at 20-40°S and 0.1-0.01 hPa showing the outflow of W2 EP        |
| 263 | flux from the instability region. This may be also due to the larger phase speed of W2,  |
| 264 | which make W2 less vulnerable to mean wind dissipations and travel more freely           |
| 265 | when propagating upward. To better quantitatively investigate the role of barotropic     |
| 266 | and baroclinic instabilities, Figure 9 shows the barotropic and baroclinic instabilities |
| 267 | separately. We found that the barotropic instability is usually ~60-80% as strong as     |
| 268 | the baroclinic instability at middle latitudes in the summer mesosphere, where it is     |
| 269 | more effective for the amplification of QTDW. In other words, the wind vertical          |
| 270 | shears generally contribute more to the growth of QTDW, but the wind curvatures are      |
| 271 | also very important.                                                                     |

I

Figure 2 has shown that both the QTDWs and the SSW peak in the middle and 272 late January, thus Figure  $\frac{8-10}{10}$  shows the comparison between the zonal mean zonal 273 wind during January 11-30 of 2005 and 2006. The zonal wind during the SSW period 274 of 2006 shows two major differences compared with that in 2005. First, the westerly 275 wind in winter stratosphere reverses to easterly. The winter westerly reversal is one 276 key feature of major SSW, which is induced by the rapid growth of stationary 277 planetary waves and their momentum deposition to the background mean flow (Liu 278 and Roble, 2002). Second, the summer easterly wind in the mesosphere is enhanced. 279 The interhemispheric couplings during SSW period have been reported in previous 280 literatures (Karlsson et al., 2007, 2009; Körnich and Becker, 2010). We then analyzed 281 the correlation between the temporal variations of the global zonal mean zonal wind 282

| 283 | and the zonal mean temperature at 70°N and 10 hPa, which increase dramatically          |
|-----|-----------------------------------------------------------------------------------------|
| 284 | during a SSW event. The correlation coefficients are shown in Figure $911$ . The zonal  |
| 285 | wind in the summer mesosphere at middle latitude shows a significant inverse            |
| 286 | relationship with the temperature variations in the winter stratosphere. In the summer  |
| 287 | hemisphere, the zonal mean zonal wind is westward in the upper stratosphere and         |
| 288 | mesosphere; it will be enhanced when the temperature in winter stratosphere increases.  |
| 289 | The SSW is mainly caused by the rapid growth of planetary waves, which deposits         |
| 290 | energy and momentum flux to the background wind. Figure 12 shows the zonal mean         |
| 291 | circulations induced by the momentum flux of SPW1, which is calculated using the        |
| 292 | downward control principle following Haynes et al. (1991) and Lubis et al. (2016). It   |
| 293 | is clear that the SPW1 induced zonal mean circulation shows maxima in winter polar      |
| 294 | stratosphere with amplitudes of -6-7 cm/s (downward) and 7-8 m/s (northward) for        |
| 295 | vertical and meridional components, respectively. It is also clear that the SPW1        |
| 296 | induced circulations are confined to the winter hemispheres, and thus contribute little |
| 297 | to the inter-hemispheric coupling. This agrees well with the mechanism that the         |
| 298 | inter-hemispheric coupling is induced by the feedback between gravity wave breaking     |
| 299 | and zonal mean zonal wind in the mesosphere (Karlsson et al., 2009; Körnich and         |
| 300 | Becker, 2010). Figure 13 shows the differences between the meridional circulation       |
| 301 | during and before the SSW following Lubis et al. (2016), which clearly indicates an     |
| 302 | anomalous cross-equator circulation from the winter to summer mesosphere. Thus, we      |
| 303 | conclude that the zonal wind anomaly during January 2006 is most likely correlated      |
| 304 | with the SSW event.                                                                     |

| 305 | We then show how these differences result in different QTDW behaviors during                   |
|-----|------------------------------------------------------------------------------------------------|
| 306 | 2005 and 2006. The mean flow instabilities of the background wind and the critical             |
| 307 | layers of W2 and W3 are shown in Figure $\frac{1014}{10}$ . First the enhanced summer easterly |
| 308 | in the mesosphere results in stronger barotropic/baroclinic instability, which provides        |
| 309 | larger forcing for the amplification of QTDW. This results in stronger W3 amplitude            |
| 310 | during 2006 than that during 2005 (Figure 3). Besides, the stronger summer easterly            |
| 311 | in the mesosphere also sustains a critical layer for W2 during 2006 at middle latitude,        |
| 312 | which is not observed in 2005. The phase speed of planetary wave is inversely                  |
| 313 | proportional to both period and zonal wave numbers, thus the phase speed of W2 is              |
| 314 | larger than W3. The existence of W2 critical layer nearby the instability region               |
| 315 | facilitates the wave-mean flow interaction, through which the energy of mean flow is           |
| 316 | transferred to W2 (Liu et al., 2004). This results in abnormally strong W2 oscillations        |
| 317 | in 2006 than that in 2005. Gu et al. (2013b) also noted that the W3 during 2006 peaks          |
| 318 | with an extremely short period of ~42 hours (also shown by Figure 1 and 4), whereas            |
| 319 | the period of W3 during austral summer tends to be longer (~52 hours) (Palo et al.,            |
| 320 | 2007; Tunbridge et al., 2011; Yue et al., 2012). The W3 QTDW with a longer period              |
| 321 | has a slower phase speed. Figure $44-15$ shows the comparison between the critical             |
| 322 | layers of 42- and 52-hour W3 for the zonal mean state during 2006. The critical layer          |
| 323 | of the 42-hour W3 runs at the edge of the mean flow instability, which is totally              |
| 324 | surrounded by the critical layer of the 52-hour W3. Thus the 52-hour QTDW signal               |
| 325 | has already been reflected away by the critical layer before it reaches the unstable           |
| 326 | region and cannot be amplified through wave-mean flow interaction (Liu et al., 2004).          |

| 327 | Figure 10b-14b also shows that both the critical layers of W3 and W2 run across the                                                                                                                                                                               |
|-----|--------------------------------------------------------------------------------------------------------------------------------------------------------------------------------------------------------------------------------------------------------------------------|
| 328 | mean flow instabilities in winter stratospheric region, whereas there is no significant                                                                                                                                                                                  |
| 329 | positive EP flux divergence near this region (Figure $\frac{1216}{12}$ ) as that shown in the                                                                                                                                                                            |
| 330 | summer mesosphere. Positive EP flux divergence indicates the energy conversion to                                                                                                                                                                                        |
| 331 | planetary waves from mean flow instability (Liu et al., 2004).source for planetary                                                                                                                                                                                       |
| 332 | waves. Thus we conclude that the mean flow instability related to the winter westerly                                                                                                                                                                                    |
| 333 | reversal during SSW period is not as effective for the QTDW amplification as that in                                                                                                                                                                                     |
| 334 | the summer mesosphere.                                                                                                                                                                                                                                                   |
| 335 | 3.3 The nonlinear coupling between W3 and SPW1                                                                                                                                                                                                                           |
| 336 | In the TIME-GCM numerical simulations, Gu et al. (2015) found that the W2                                                                                                                                                                                                |
| 337 | peaks earlier than W3 due to the fact that the W2 has a larger phase speed and thus                                                                                                                                                                                      |
| 338 | suffers weaker dissipation during its propagation and amplfication. We should also                                                                                                                                                                                       |
| 339 | note that the W2 is emmediately genearted through the nonlinear interaction, when                                                                                                                                                                                        |
| 340 | the W3 and SPW1 are forced simutaneously at the lower model boundary. However,                                                                                                                                                                                           |
| 341 | we found that the W2 peaks later than W3 duirng January 2006, which suggest a later                                                                                                                                                                                      |
| 342 | occurrence of the nonlinear interaction. Gu et al. (2015) proposed that the nonlinear                                                                                                                                                                                    |
| 343 | interaction between W3 and SPW1 could also provide sources for W2. We also                                                                                                                                                                                               |
| 344 | calculated the nonlinear advection between W3 and SPW1 following Gu et al. (2016c)                                                                                                                                                                                       |
| 345 | as a substitute to represent their nonlinear interaction:                                                                                                                                                                                                                |
| 346 | $F_{nonlinear,y} = -\frac{1}{a\cos\varphi} \left( u_1 \frac{\partial v_2}{\partial \lambda} + u_2 \frac{\partial v_1}{\partial \lambda} \right) - \frac{1}{a} \left( v_1 \frac{\partial v_2}{\partial \varphi} + v_2 \frac{\partial v_1}{\partial \varphi} \right) $ (3) |
| 347 | The meridional nonlinear advection which is shown in Figure 1317. The nonlinear                                                                                                                                                                                          |

[revised manuscript text omitted]

This work is sponsored by the Project Funded by China Postdoctoral Science 455 Foundation (2015M582001, 2016T90573), the National Natural Science Foundation 456 of China (41421063, 41304123), and Hundred Talents Program (D). The 457 NOGAPS-ALPHA dataset is available at ftp://map.nrl.navy.mil/pub/nrl/nogaps and 458 downloaded 459 the Aura/MLS temperature observation can be by

| 460 | https://acdisc.gesdisc.eosdis.nasa.gov/data/Aura_MLS_Level2/.https://disc.sci.gsfc.na |
|-----|---------------------------------------------------------------------------------------|
| 461 | sa.gov/Aura/data-holdings/MLS.                                                        |
| 462 |                                                                                       |

**463 **References**

[revised manuscript text omitted]